# Cell diversity and plasticity during atrioventricular heart valve EMTs

Jeremy Lotto [1,2], Rebecca Cullum[1], Sibyl Drissler[1,2], Martin Arostegui [3,4], Victoria C. Garside [1,2,5], Bettina M. Fuglerud[1,6], Makenna Clement-Ranney [1], Avinash Thakur[1,6], T. Michael Underhill [3,4,7] & Pamela A. Hoodless [1,2,6,7] ✉

Epithelial-to-mesenchymal transitions (EMTs) of both endocardium and epicardium guide atrioventricular heart valve formation, but the cellular complexity and small scale of this tissue have restricted analyses. To circumvent these issues, we analyzed over 50,000 murine single-cell transcriptomes from embryonic day (E)7.75 hearts to E12.5 atrioventricular canals. We delineate mesenchymal and endocardial bifurcation during endocardial EMT, identify a distinct, transdifferentiating epicardial population during epicardial EMT, and reveal the activation of epithelial-mesenchymal plasticity during both processes. In *Sox9*-deficient valves, we observe increased epithelial-mesenchymal plasticity, indicating a role for SOX9 in promoting endothelial and mesenchymal cell fate decisions. Lastly, we deconvolve cell interactions guiding the initiation and progression of cardiac valve EMTs. Overall, these data reveal mechanisms of emergence of mesenchyme from endocardium or epicardium at single-cell resolution and will serve as an atlas of EMT initiation and progression with broad implications in regenerative medicine and cancer biology.

The atrioventricular (AV) valves are unique structures composed of three extracellular matrix (ECM) layers ensheathed by valve endocardial cells (VECs) and populated by valve interstitial cells (VICs)[1]. VICs, which are specialized cardiac fibroblasts, are generally quiescent under homoeostatic conditions, and their activation is intimately associated with valve disease states, including calcification and regurgitation[2,3]. It is thought that this pathogenic, activated state may recapitulate developmental processes; however, few specific parallels have been elucidated due to our limited understanding of valvulogenesis.

During mouse embryogenesis, AV mesenchyme, which comprises the progenitor pool for VICs, is formed early in development through two epithelial-to-mesenchymal transitions (EMTs) from distinct cellular sources: endocardium and epicardium[4–6] (Fig. 1a). Around E8.5, a constriction forms at the junction between the atria and ventricles,

where myocardial hyaluronan and fibronectin accumulate, establishing the primitive valves, called the AV cushions[4,7]. Around this stage, the early heart also begins to beat, increasing the hemodynamic load exerted on the endocardium[8]. By E9.0, fluid shear stress from blood flow and signals from the myocardium, including BMPs, induce an endocardial-to-mesenchymal transition (EndMT) in a subset of endocardial cells lining the cushions[9–13]. These cells lose their endothelial cell-cell contacts, invade the ECM within the AV cushions, and adopt a mesenchymal phenotype (Fig. 1a). This transdifferentiation event is controlled by various transcription factors, like SNAI2[14], GATA4[15], and SOX9[16,17], which are activated in response to cues from neighbouring cell types, including the TGFβ superfamily[9,10], WNT[18], and NOTCH[19,20]. VEGF signalling is also implicated – inducing NFATc1 expression, which represses EndMT and maintains endothelial identity in non-transitioning endocardium[21].

[1]Terry Fox Laboratory, BC Cancer, Vancouver, BC, Canada. [2]Cell and Developmental Biology Program, University of British Columbia, Vancouver, BC, Canada. [3]Biomedical Research Centre, University of British Columbia, Vancouver, BC, Canada. [4]Department of Cellular and Physiological Sciences, University of British Columbia, Vancouver, BC, Canada. [5]Department of Anatomy and Physiology, University of Melbourne, Melbourne, VIC, Australia. [6]Department of Medical Genetics, University of British Columbia, Vancouver, BC, Canada. [7]School of Biomedical Engineering, University of British Columbia, Vancouver, BC, Canada. ✉e-mail: hoodless@bccrc.ca

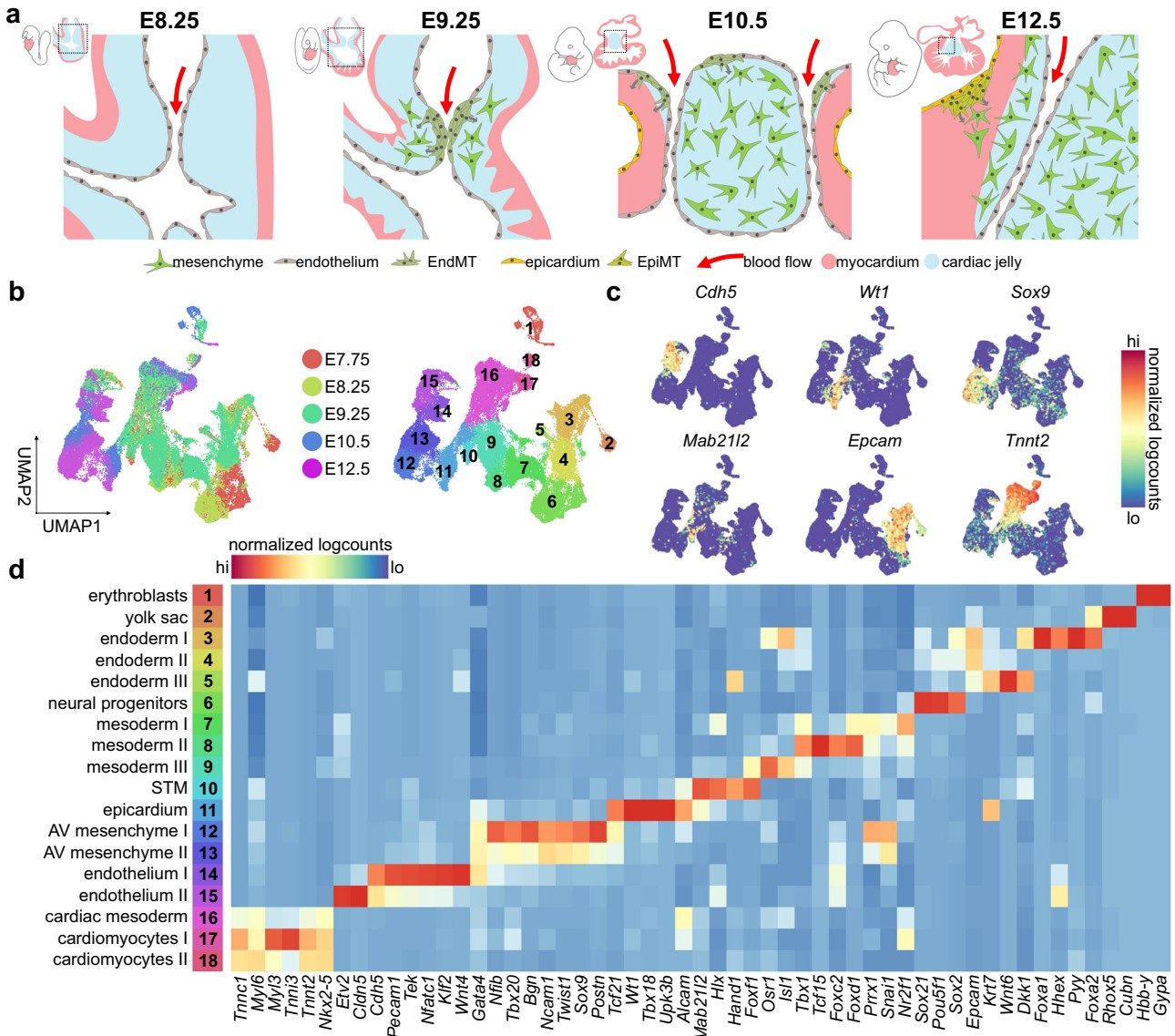

**Fig. 1 | Tracking the emergent lineages of the heart and early atrioventricular canal. a** Overview of murine atrioventricular development, highlighting the endocardial-to-mesenchymal transition (EndMT) at E9.25-10.5 and the epicardial-to-mesenchymal transition (EpiMT) at E12.5. **b** UMAP visualization of 48,822 cells from E7.75-E9.25 hearts and E10.5-E12.5 AV canals. Cell types are denoted by colour and number in legend in panel **d. c** UMAP visualization of select gene expression for endothelial, epicardial, mesenchymal, and myocardial biomarkers. **d** Select differentially expressed genes in complete dataset, showing emergence of endothelial, epicardial, myocardial, and mesenchymal populations. AV atrioventricular, STM septum transversum mesenchyme. See also Supplementary Fig. 1 and Supplementary Data 1–2. Source Data are provided as a Source Data file.

AV mesenchymal cells are also generated from the epicardium via the epicardial-to-mesenchymal transition (EpiMT) starting around E12.5 (Fig. 1a). The epicardium is the cardiac mesothelium, derived from septum transversum mesenchyme (STM), from which a subset of cells extend across the pericardial space to populate the surface of the heart[22]. Via EpiMT, epicardial cells contribute to smooth muscle cells, vasculature, pericytes, and fibroblasts, including those of the atrioventricular canal (AVC) and the annulus fibrosus[5,6,23,24]. However, the processes underlying AV EpiMT remain poorly characterized, as the few studies assessing EpiMT focus on chamber epicardium[25–29].

Once EndMT and EpiMT are complete, AV mesenchymal cells drive AV septation, delamination of the cushions, and remodelling to form the three laminae of ECM. Maturation of valve cell lineages also continues postnatally[30]. While our understanding of atrioventricular development has improved greatly with the advent of genomics research, the details

underlying many aspects of AV EMTs are incompletely characterized, mainly due to limitations related to the size of the embryonic valves. These include the cellular diversity present in this tissue during EMT; the mechanisms of initiation, progression, and resolution of these processes; and the comparison of EMT of endocardial and epicardial origin that contribute to AV mesenchyme. Mechanisms underlying the formation of the valves is pertinent to understanding the aetiology of congenital defects and disease, including cardiac fibrosis and valve disease, where EndMT genetic programmes may reactivate to drive pathogenesis[31,32]. Of note, defining details controlling EMT initiation and progression may further elucidate mechanisms that drive cancer metastasis as significant parallels exist between these processes[33]. In this work, we apply scRNA-seq to reveal mechanisms of emergence of mesenchyme from endocardium and epicardium and provide histological data to show the spatiotemporal dynamics during these processes.

## Results

### Single-cell transcriptomics captures early valvulogenesis

Single-cell RNA sequencing (scRNA-seq) has been successfully applied to chart transcriptional diversity within the early embryo[34,35], heart[36], liver[37,38], and others. In the present study, we generated five scRNA-seq libraries from E10.5 and E12.5 AVCs and complemented these datasets with previously published E7.75, E8.25, and E9.25 heart datasets from the Srivastava group[36] (Fig. 1b–d and Supplementary Fig. 1a–j). Our analyses cover the earliest stages of AV development, including AV endocardial specification at E8.25, EndMT induction and progression from E9.25-E10.5, and EpiMT initiation at E12.5. In addition, we generated data from endothelial-specific *Sox9* conditional knockout AVCs to compare with normal development (Supplementary Fig. 1a).

A total of 53,521 single-cells from all datasets passed stringent quality control checks, with a median of 4657 genes detected per cell (Supplementary Fig. 1i, j and Supplementary Data 1). We used PhenoGraph[39] clustering to identify diversity captured within the 48,822 wildtype cells and annotated 18 distinct populations, including emergent mesenchymal, myocardial, and endothelial populations (Fig. 1b–d and Supplementary Data 2). Endocardial- and epicardial-derived lineages were computationally segregated from each timepoint and analyzed independently (Supplementary Fig. 1b–f). Together, these data define the earliest stages of AV valve formation, including the acquisition and maintenance of mesenchymal, endocardial, and epicardial identities, and provide insight into the mechanisms underlying EMTs in vivo.

### Activation of epithelial-mesenchymal plasticity during EndMT

Around E8.5, the accumulation of cardiac jelly at the AVC forms the AV cushions[4] (Fig. 1a). Initially acellular, these structures are colonized by endocardial cells as they undergo EndMT. By E9.5, these cells lose cell-cell contacts, lose apico-basal polarity, gain migratory and invasive characteristics, and transdifferentiate to form AV mesenchyme, which is essential for establishing valve architecture[1,4] (Fig. 1a). To characterize this process, we analyzed 10,354 single-cell transcriptomes from E7.75-12.5, and performed batch correction using Harmony, which approximates consecutive timepoints in a timeseries[34]. Clustering identified eight populations including early endothelium, endocardium, AV endocardium, two VEC progenitor populations, and three AV mesenchymal clusters (Fig. 2a, Supplementary Fig. 2a–c, and Supplementary Data 2). Some transcriptional differences across batches persist after correction, though these genes are predominantly associated with general biological processes and metabolism (Supplementary Fig. 2a and Supplementary Data 3).

Cells within endocardial, AV endocardial, and VEC clusters express various endothelial biomarkers, including *Pecam1*, *Tek*, and *Cdh5*. The AV endocardial cluster, composed predominantly of cells from E9.25, has activated genes associated with AV identity, such as *Gata4* (Fig. 2b). Endocardial *Gata4* is required for EndMT initiation[15], and immunostaining shows that GATA4 is enriched within the endocardial and endocardial-derived cells of the AVC compared to the ventricles from E9.5 until E12.5 (Supplementary Fig. 2d, e). This strongly supports its function in the establishment of early AV identity. Two distinct VEC populations emerge in the AVC by E12.5, including VECs I, which expresses *Ednrb*, *Irx5*, and *Palmd*, and VECs II, which is enriched for *Wnt4*, *Nfatc1*, and *Edn1*. The three AV mesenchymal populations all express regulators of mesenchymal identity, such as *Sox9*[16], *Msx1*[40], and *Tbx20*[41], but differ in timepoint composition or proliferative status (Fig. 2b and Supplementary Fig. 2b). Researchers have previously proposed a distinct proliferative zone of AV mesenchyme exists within the cushion[42,43] and our transcriptional data suggest a proliferative subset emerges by E12.5. However, immunostaining for phospho-histone H3 (pH3) or PCNA does not reveal any obvious patterns of proliferation within the cushions, suggesting proliferation is not spatially coordinated, but is activated ubiquitously

in the emerging mesenchyme at E12.5 (Supplementary Fig. 2f–h). Overall, these analyses define expression signatures for distinct populations of endocardial and mesenchymal cells in early AV valves.

We next applied Palantir to infer developmental trajectories, including differentiation and cell fate choice, from our data to characterize the divergence between VEC progenitors and the AV mesenchyme during EndMT[44] (Fig. 2c–e). Palantir orders cells during a process along pseudotime, then for each cell, assigns a probability for differentiating towards terminal cell states. Differentiation potential, a measure of cell plasticity, is calculated using these probabilities; multipotent cells have the highest differentiation potential and mature terminal states have the lowest potential. An early endothelial *Etv2*hi start cell was selected from which trajectories were calculated to either VEC progenitor or AV mesenchymal termini (Fig. 2c and Supplementary Fig. 3i–k). Differentiation potential is high among the early endothelial cluster, but highest where endocardial and mesenchymal lineages begin to bifurcate (arrows, Fig. 2c). Differentiation potential toward the AV mesenchymal terminus over pseudotime shows that a portion of cells within the AV endocardial and VEC II clusters by E9.25 and E10.5 are primed toward an AV mesenchymal identity (Fig. 2e). By E12.5, most cells have segregated into the endocardial and mesenchymal lineages as AV EndMT is complete.

Gene trends along pseudotime were calculated to characterize the divergence between these terminal states and showed the dynamic regulation of factors during this process (Fig. 2d). In the AV mesenchymal lineage, expression of *Gata4* is retained, but endothelial biomarkers are repressed. Instead, we see the activation of various transcription factors, including master regulators of AV mesenchymal identity, such as *Msx1*, *Sox9*, and *Tbx20*[1], along with factors not assessed during EndMT, like *Nfib* and *Klf1* (Fig. 2d). VEC progenitors retain expression of endothelial biomarkers and further activate expression of a variety of genes associated with AV and VEC identity, including *Gata4*[15], *Irx5*[45], *Nfatc1*[21], and *Wnt4*[46], along with genes with uncharacterized roles in AV development, like *Ednrb* and *Palmd* (Fig. 2d). Interestingly, *Palmd* has been identified as a susceptibility gene in calcific valve disease[47]. Immunostaining illustrates this cell fate decision. At E10.5, we observe NFATc1 co-expression on the flow-side of the cushions with ERG and within the VECs lining the primitive valves at E11.5[21,48] (Fig. 2f). Overall, these data detail bifurcation of endocardial and mesenchymal cell lineages and previously unappreciated diversity during AV EndMT.

### Emergence of epithelial-mesenchymal plasticity during EndMT

Remarkably, we observe appreciable overlap of endocardial and mesenchymal factors during EndMT, including expression of *Pecam1*, *Tek*, *Twist2*, and *Msx1* in our scRNA-seq data (Fig. 2d) and co-localization of SOX9, ERG, and NFATc1 by histology (arrowheads, Fig. 2f). To probe this shift in phenotype during EndMT, we calculated average gene expression profiles for each cell from a list of known endothelial or AV mesenchymal biomarkers (Supplementary Data 4). We further calculated a fluid shear stress response profile from genes identified to assess gene activation in response to hemodynamic forces (See Methods; Supplementary Data 4). As expected, the endothelial gene profile is highest within endocardial and VEC populations, while the mesenchymal profile is highest among emerging AV mesenchyme (Fig. 3a). However, we observe co-expression of mesenchymal and endothelial gene profiles within AV endocardial and VEC II clusters during EndMT (arrows, Fig. 3a, b and Supplementary Fig. 3a). The gene profile representing fluid shear stress response is highest among AV endocardial and VEC II clusters in the same domain where epithelial and mesenchymal gene profiles are co-expressed (Fig. 3a and Supplementary Fig. 3b). Of all characterized mechanosensitive ion channels, *Piezo1* is the most highly expressed in endocardium, suggesting it acts as the major mechanosensor sensing fluid shear stress during EndMT in mouse, as is observed in zebrafish[13,49]

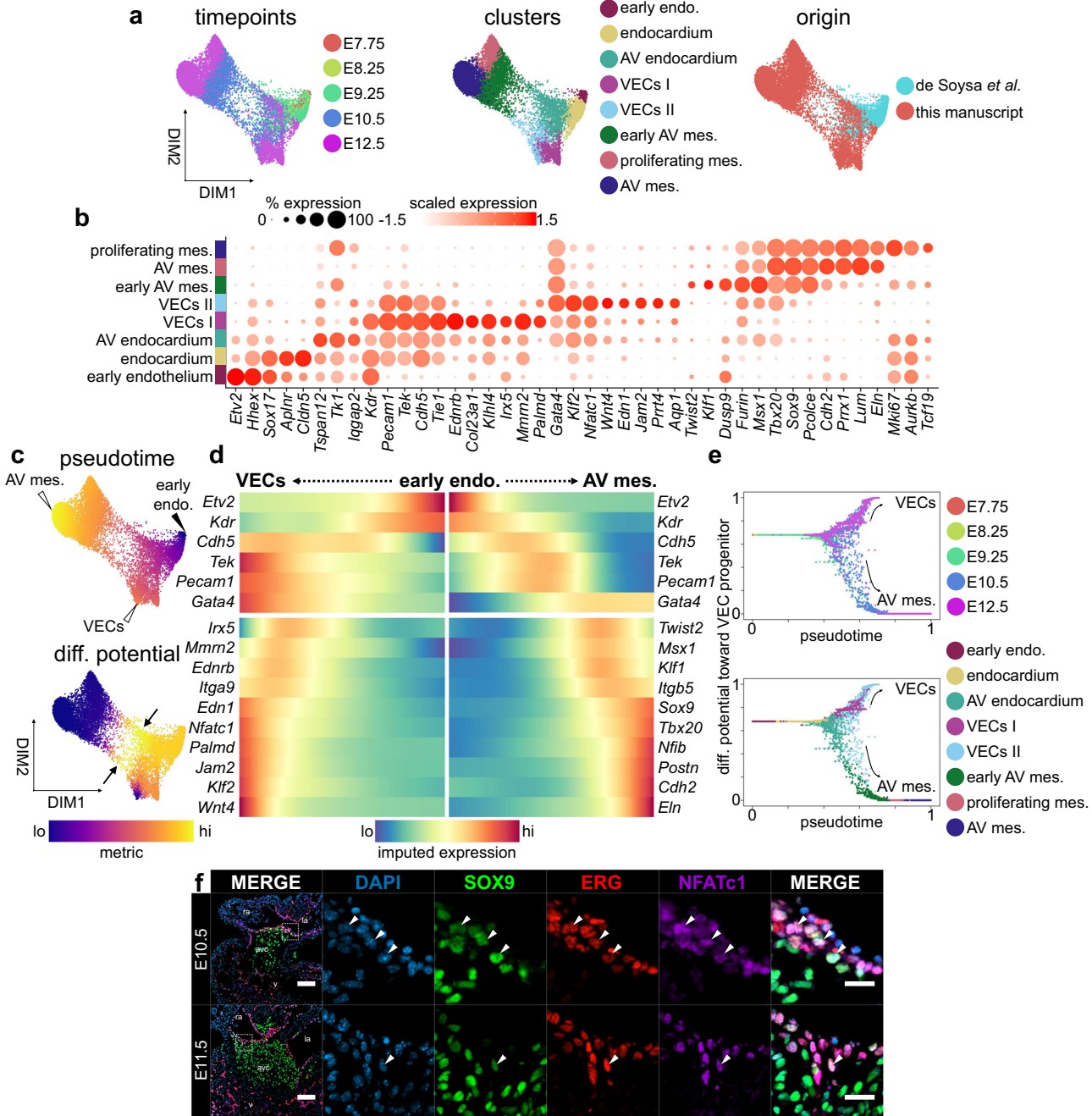

**Fig. 2 | Endocardial and mesenchymal cell lineage bifurcation during EndMT.**
**a** Force-directed layouts of 10,354 endocardial and endocardial-derived cells displaying timepoint, annotated cluster, and origin. **b** A dot plot displaying select genes enriched within endothelial-derived clusters, including VEC populations and AV mesenchyme. Size of the dot represents proportion of the cluster that expresses each gene. Colour indicates level of expression. **c** Force-directed layouts of 10,354 endocardial and endocardial-derived cells displaying pseudotime and differentiation (diff.) potential. Arrows indicate high diff. potential in domains of endothelial-mesenchymal plasticity (EMP). **d** Gene trends along pseudotime show bifurcation

between VEC and AV mesenchymal trajectories during the endocardial-to-mesenchymal transition. **e** Differentiation potential as a function of pseudotime for endothelial-derived lineages progressing toward a VEC terminus. Each dot represents a single-cell that is colour-coded by embryonic timepoint or cluster. **f** Immunostaining of AV canals for SOX9, ERG, and NFATc1 shows co-expression of on the flow side of the valves (white arrowheads) at E10.5 (n = 10) and E11.5 (n = 4). Scale bars: 100 μm, 25 μm in regions of interest (ROIs). AV atrioventricular, endo. endothelium, mes. mesenchyme. See also Supplementary Fig. 2 and Supplementary Data 2.

(Supplementary Fig. 3c). These findings strongly suggest the development of a cell state with epithelial-mesenchymal plasticity (EMP) during EndMT in domains of high fluid shear stress.

Histology confirms the presence of an EMP hybrid cell state on the flow-side of the cushions where the hemodynamic load is highest[50]. Here, we see co-expression of mesenchymal PDGFRα and SOX9 with the endothelial biomarkers CD31 and ERG in cells at E9.25 and E10.5 (arrowheads, Fig. 3c, d and Supplementary Fig. 3d–f). Co-expression is

significantly reduced once mesenchymal and endothelial lineages have resolved at E12.5 (Fig. 3c, d and Supplementary Fig. 3d–f). NFATc1⁺ cells are also found within these domains of EMP, indicating that VEC progenitors also emerge from this population (Fig. 2f). This is supported by our single-cell data, where we observe overlap between cells exhibiting EMP and high differentiation potential toward both VEC and AV mesenchymal termini, suggesting bipotency (Figs. 2c and 3a). Together, these findings indicate that cells displaying EMP emerge during

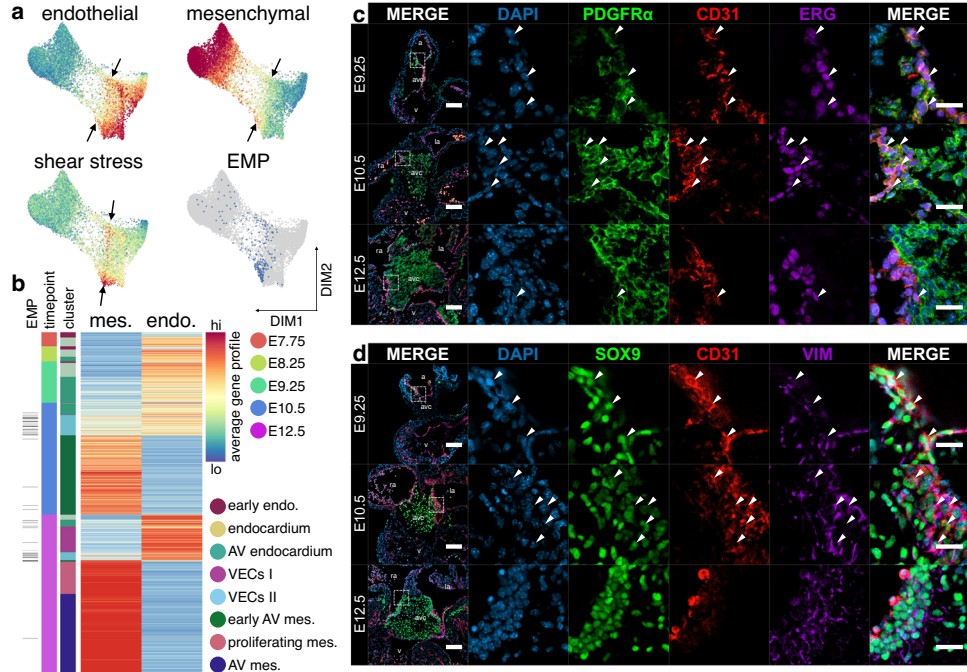

**Fig. 3 | Activation of epithelial-mesenchymal plasticity during EndMT. a** Force-directed layouts of 10,354 endocardial and endocardial-derived cells displaying average gene profiles denoting expression of endothelial and mesenchymal bio-markers as well as transcriptional response to shear stress, highlighting domains of EMP or high fluid shear stress response (arrows). **b** Heatmaps displaying average endothelial and mesenchymal gene profiles for endocardial and endocardial-derived cells from E7.75-12.5, where overlap indicates emergence of EMP.

Immunostaining of AV canals for (**c**) PDGFRα, CD31, and ERG or (**d**) SOX9, CD31, and vimentin show widespread co-expression of endothelial and mesenchymal biomarkers, primarily at E9.25 and E10.5, suggesting EMP (arrowheads). Results are representative of three to nine independent experiments. Scale bars: 100 μm, 25 μm in ROIs. AV atrioventricular, endo endothelium, mes mesenchyme, VECs valve endocardial cells. See also Supplementary Fig. 3 and Supplementary Data 2 and 4.

early valvulogenesis, likely in response to the flow of blood. Our analyses predict that these hybrid cells exhibit high cell lineage plasticity and may be a common source for both VECs and AV mesenchyme.

**Increased epithelial-mesenchymal plasticity in Sox9 cKO AVCs**

SOX9 is critical for AV valve development, including in EndMT and matrix remodelling[16,17,51,52], and its deregulation or mutation is associated with calcification and campomelic dysplasia, in which valve defects are often present[53–55]. In mouse, the role of *Sox9* in AV valve development has been studied using *Sox9fl/fl;Tie2-cre* (*Tek*-cre) embryos, where *Sox9* is ablated in endothelial-derived populations, including AV mesenchyme derived from EndMT[56]. These embryos die around E12.5 with AV developmental defects, characterized by severe cushion hypoplasia[17]. Despite its known roles in promoting AV valve development, surprisingly little is known about *Sox9*'s molecular mechanisms during valve development beyond what can be seen in the terminal phenotypes.

To address this gap in knowledge we generated scRNA-seq libraries from *Sox9fl/fl;Tie2-cre* (*Sox9* cKO) AVCs along with *Sox9fl/+;Tie2-cre* or *Sox9+/+;Tie2-cre* normal littermate controls at E10.5 (Supplementary Fig. 1a and 4a–c). Analysis of *Sox9* cKO and WT littermates shows a significant overlap between epicardial, endocardial, hematopoietic, and cardiomyocyte clusters (Supplementary Fig. 4a, c). However, pronounced differences exist in the distribution of mesenchymal clusters, where cKO and WT cells almost completely segregate (Supplementary Fig. 4a, c).

To assess shifts in cell identity between the WT and *Sox9* cKO lineages, we computationally segregated endocardial lineages of the cKO library and overlaid them on the complete endocardial lineage force-directed layout to compare with our E10.5 WT EndMT data (Fig. 4a–c). Differential gene expression analysis between WT and *Sox9* cKO cells in AV endocardial and VEC clusters shows modest

differences in expression of endothelial biomarkers and factors (Fig. 4a). Within mesenchymal populations, however, we observe striking changes in the transcriptomes between WT and *Sox9* cKO cells (Fig. 4a and Supplementary Fig. 4d; Supplementary Data 2). In WT mesenchyme we see enrichment of transcription factors, including *Klf1*, *Scx*, and *SoxS*, signalling components, such as *Fgfr2*, *Notch2*, *Tgfbi*, and *Pdgfra*, and ECM components, like type II and IX collagens, *Fbln2*, and *Matn4* (Fig. 4a, Supplementary Fig. 4d, and Supplementary Data 2). Of note, we observe the enrichment of integrin receptor components in WT mesenchyme, including *Itga4* and *Itgb5*, which are required for cell adhesion to FN—a major cushion ECM component at this stage[7,57] (Fig. 4a). Conversely, in mesenchyme sequenced from *Sox9* cKO AVCs, we see stronger enrichment of a variety of master regulators of EndMT and mesenchymal development, including *Twist1*, *Snai2*, *Msx2*, and *Zeb2* along with cell signalling components and ECM genes, including *Tagln*, *Pdgfrb*, *Angpt1*, *Tnc*, *Postn*, *Bmp4*, *Flrt2*, as well as type I, V, and XIII collagens (Fig. 4a, Supplementary Fig. 4d, and Supplementary Data 2). BMP signalling repressors, such as *Fst* and *Smad7* are also retained within cKO mesenchyme, which could affect EndMT progression[10,58]. Gene ontology analysis of enriched genes in WT and *Sox9* cKO mesenchyme shows prominent overlap between mesenchymal and developmental ontologies; however, ontologies associated with cardiac muscle development are specifically enriched within the cKO (Supplementary Fig. 4e). This is reflected in the activation of myosin and actin genes in the cKO mesenchyme, including *Myh10*, *Myl9*, *Tpm1*, *Acta2*, and *Actc1*, suggesting the adoption of a more myofibroblastic identity (Fig. 4a and Supplementary Fig. 4d). These findings suggest that without *Sox9*, we see a significant shift in mesenchymal cell identity in the AVC, specifically in cell adhesion, matrix production, and cell signalling.

Profound changes related to cell identity, including shifts in mesenchymal and endothelial phenotype, are observed among

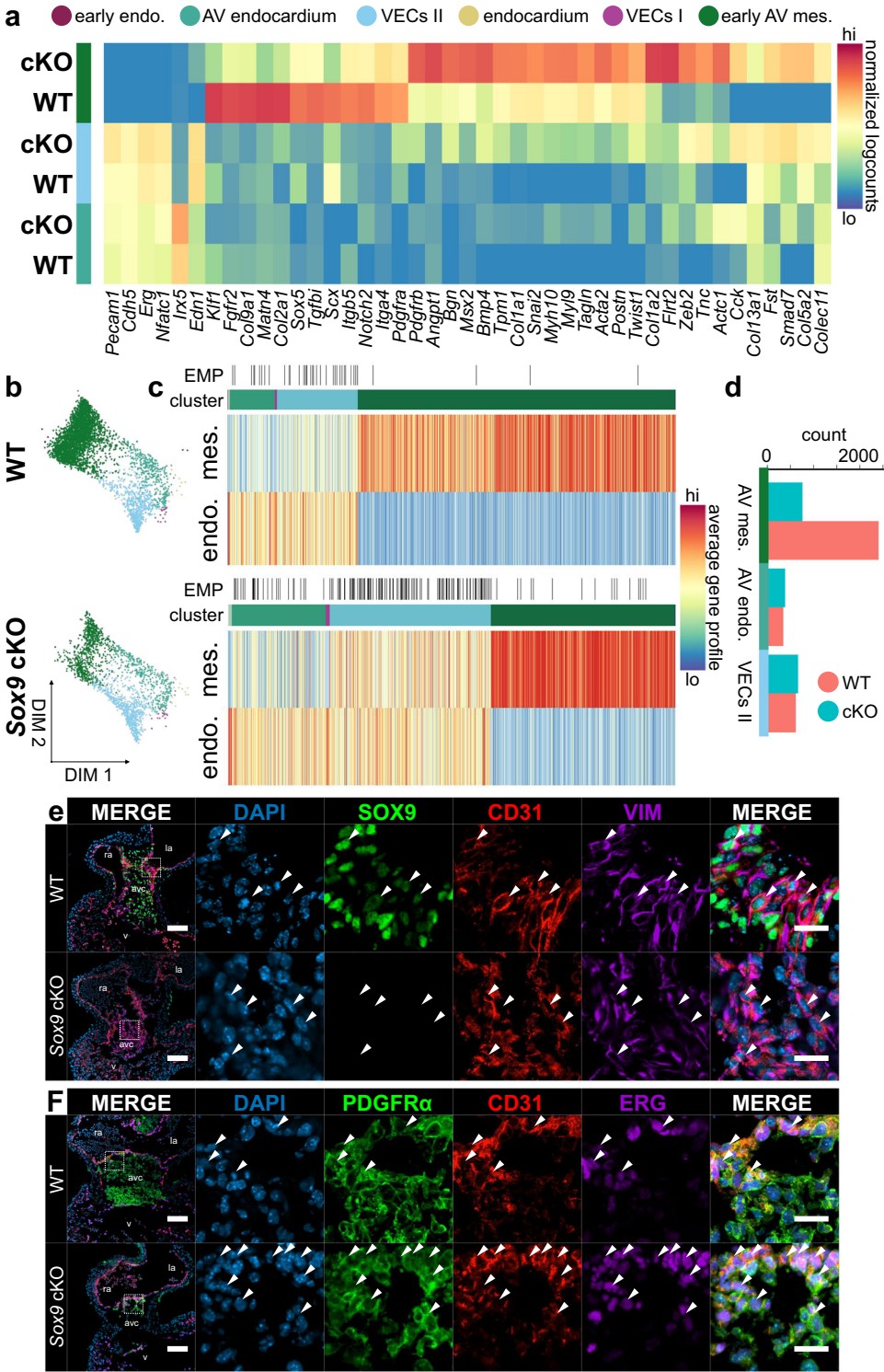

**Fig. 4 | Perturbations of endothelial and mesenchymal cell states in *Sox9* conditional knockout AV canals. a** In the absence of *Sox9*, differential gene expression analysis reveals profound differences in identity between emergent mesenchymal populations as well as increased expression of mesenchymal factors in VECs. **b** Force-directed layouts of 3368 WT and 1810 *Sox9* cKO endocardial and endocardial-derived cells from E10.5 displaying annotated cluster contribution. **c** Heatmaps displaying average endothelial and mesenchymal gene profiles for E10.5 WT and *Sox9* cKO endocardial and endocardial-derived cells, showing a profound increase in cells exhibiting EMP in the absence of *Sox9*. **d** Cell distribution in the three main clusters captured at E10.5 in the WT and *Sox9* cKO endocardial-

derived lineages, showing reduced numbers of mesenchymal cells in the cKO. Comparing (**e**) SOX9, CD31, and vimentin or (**f**) PDGFRα, CD31, and ERG immunostaining in E10.5 *Sox9* cKO and WT AV canals reveals developmental arrest of endocardial cells with mesenchymal features, like VIM or PDGFRα (white arrowheads). Few fully transdifferentiated mesenchymal cells lacking CD31 expression are observed in the *Sox9* cKO. Results are representative of three to six independent experiments. Scale bars: 100 μm, 25 μm in ROIs. AV atrioventricular, endo. endothelium, mes mesenchyme. See also Supplementary Fig. 4 and Supplementary Data 2 and 4. Source Data are provided as a Source Data file.

endothelial-derived cell lineages in the *Sox9* cKO (Fig. 4b, c and Supplementary Fig. 4f, g). This includes increased proportions of AV endocardial and VEC II cells and reduced numbers of AV mesenchymal cells (Fig. 4b–d and Supplementary Fig. 4g). Further, we see a four-fold increase of cells that exhibit robust EMP (Fig. 4c and Supplementary Fig. 4f, g). Within the *Sox9* cKO, some mesenchymal cells do emerge. Contradictorily, they display an increased mesenchymal gene profile, in part due to increased expression of EMT transcription factors, like *Snai1*, *Twist1*, and *Zeb2*[i] (Fig. 4a, c and Supplementary Fig. 4f, g). These results, including the increased proportions of cells exhibiting EMP and the upregulation of other EMT factors, reveal profound dysregulation of cell type identity in the *Sox9* cKO.

To validate our findings in the *Sox9* cKO, we immunostained E10.5 *Sox9*[fl/fl];*Tie2-cre* hearts for several factors associated with EndMT and AV identity (Fig. 4e, f and Supplementary Fig. 4h-k). Our results show decreased proportions of mesenchymal cells within the cKO AVC, in line with decreased numbers of captured mesenchymal cells in the *Sox9* cKO single-cell data (Fig. 4b–d Supplementary Fig. 4j–l). Expression of ERG, an endothelial master regulator, and CD31 (*Pecam1*), an endothelial cell adhesion molecule, is retained in cells at increased proportions within the AVC, indicating a failure of these cells to transdifferentiate to mesenchyme (Supplementary Fig. 4h, i). Immunostaining suggests maintenance of AV identity and promotion of VEC identity in the *Sox9* cKO, where increased proportions of GATA4+ NFATc1+ endocardial cells are observed (Supplementary Fig. 4h-i). Increased numbers of cells exhibiting EMP at E10.5 are also observed by immunostaining (Fig. 4e, f and Supplementary Fig. 4l). We observe increased numbers and proportions of cells co-expressing mesenchymal PDGFRα and the endothelial biomarker CD31 at E10.5 in the *Sox9* cKO (arrowheads, Fig. 4e, f and Supplementary Fig. 4l). Flow cytometry and immunostaining of sorted single-cells from individual E10.5 hearts further support increased proportions of endothelium (CD31+ PDGFRα–) and cells exhibiting EMP (CD31+ PDGFRα+) in the cKO (Supplementary Fig. 4m–o). Of note, differences in proliferation between WT and *Sox9* cKO mesenchyme are observed at E9.5 and E12.5, suggesting that the phenotype observed in the *Sox9* cKO may be multifactorial (Supplementary Fig. 4p–r). Overall, our data suggest that EndMT stalls without *Sox9*, leading to the developmental arrest of endocardium exhibiting EMP, while the few cells that do transdifferentiate to mesenchyme adopt a more myofibroblastic phenotype.

## Hic1 marks a distinct epicardial cell type during EpiMT

At E12.5, EpiMT is initiated in epicardial cells lining the outside of the AVC[5,59]. These cells gain migratory and invasive characteristics, transdifferentiate to mesenchyme, and populate the right and left lateral cardiac cushions, along with the annulus fibrosis[5,6,23,59]. While chamber epicardium development has been described using single-cell transcriptomics[27,60], lineage divergence of epicardial and mesenchymal populations during AV EpiMT remains uncharacterized. To detail this bifurcation, we analyzed epicardial progenitors along with a portion of the AV mesenchymal lineages captured within our datasets from E9.25 to E12.5 (Supplementary Fig. 1d–f). In total, we analyzed 4,633 single-cell transcriptomes, which correspond to five populations including septum transversum mesenchyme (STM), three epicardial populations, and AV mesenchyme (Fig. 5a and Supplementary Fig. 5a–c). Of note, epicardial lineage contribution to AV mesenchymal cells, without lineage tracing, could not be confirmed. In this analysis, the inclusion of AV mesenchyme acts as an anchor point, providing developmental context for us to investigate the process of EpiMT.

Within the STM, we see moderate activation of characterized regulators of epicardial identity, including *Tbx18*, *Wt1*, and *Lhx2* (Fig. 5b). Early epicardium and epicardium differ slightly in the expression of epicardial biomarkers, including expression of *Lhx9* (Fig. 5b). We further identify an AV epicardial-derived cell (AV EPDC) population. While these cells retain expression of some epicardial

transcription factors, including low *Wt1* and *Tbx18*, and robust *Tcf21*, they are also enriched for genes typically associated with AV mesenchymal identity, including *Postn*, *Itgb5*, and *Lum*. In addition, they express a variety of genes uncharacterized in this context, including *Col6a2*, *Pid1*, *Islr*, and *Hic1*, suggesting these cells have progressed towards mesenchyme (Fig. 5b).

We applied trajectory analysis to characterize the divergence between the epicardium and the AV mesenchyme during EpiMT. An early E9.25 STM start cell was selected from which trajectories were calculated to either epicardial or AV mesenchymal termini (Fig. 5c and Supplementary Fig. 5d, e). Differentiation potential is high among the STM populations, but is highest at the interface between the epicardial and AV EPDC clusters (arrows, Fig. 5c). This suggests that AV EPDCs are specified from the epicardium and not directly from a mesodermal progenitor earlier in development. Differentiation potential toward the AV mesenchymal terminus over pseudotime shows that by E10.5 a small subset of cells is primed towards an AV mesenchymal cell fate (Supplementary Fig. 5f). In line with this observation, we see activation of *Hic1* and *Itgb5*, markers of AV EPDCs, in some epicardial cells as early as E10.5 (Supplementary Fig. 5g). This suggests that EpiMT may initiate by E10.5, which is significantly earlier than previously understood.

Gene trends along pseudotime were calculated to better characterize the divergence between these terminal states (Fig. 5d). We see that for both epicardial and AV mesenchymal trajectories, repression of septum transversumnal biomarkers, such as *Alcam*, *Foxf1*, and *Mab21l2* is coincident with the activation of transcription factors associated with the emergent epicardium, like *Lhx9* and *Wt1* (Fig. 5d). Cells fated toward an epicardial terminus maintain expression of various transcription factors associated with epicardial identity, including *Wt1*, *Tbx18*, and *Tcf21*. Biomarkers of embryonic epicardium are also activated at this stage, including *Lrrn4*, *Msln*, *Apela*, and the uroplakins (Fig. 5d). Conversely, in cells fated towards an AV mesenchymal terminus, factors associated with epicardial identity are maintained at low levels. *Tcf21* is the sole canonical epicardial transcription factor that is not repressed as AV EPDCs emerge. Of note, in mice lacking *Tcf21*, chamber EpiMT is impaired[61]. Along with retained *Tcf21*, AV EPDCs express *Hic1* and *Itgb5* before induction of a variety of master regulators of AV mesenchymal identity, including *Sox9* and *Tbx20* (Fig. 5d). These data capture AV EpiMT and reveal the dynamic regulation of transcription factors, cell adhesion molecules, and signalling components. Interestingly, we also identify a distinct population of *Hic1*-expressing EPDCs from chamber epicardium data from Quijada et al.[27]. Comparing AV and chamber EPDC populations reveal transcriptional differences likely related to the fate of the transdifferentiating populations, including enriched *Tagln*, *Acta2*, and *Twist1* within AV EPDCs and *Igf2*, *Apela*, and *Upk3b* within chamber EPDCs (Supplementary Fig. 5h, i).

To probe shifts in phenotype during EpiMT, we calculated average gene expression profiles for each cell from a list of known epicardial or AV mesenchymal biomarkers (Supplementary Data 4). Consistent with observations during EndMT, we see induction of EMP in domains of high differentiation potential at E10.5 and E12.5 (Fig. 6a, b and Supplementary Fig. 6a). This is the result of epicardial gene activation along with moderate expression of genes associated with mesenchymal identity at the onset of EpiMT (Fig. 6a, b and Supplementary Fig. 6a). This includes *Cdh2*, *Snai1*, and *Col1a1* among others (Supplementary Fig. 6b). These mesenchymal factors are further activated, and epicardial factors are further reduced, as cells transdifferentiate to AV mesenchyme via an AV EPDC intermediate during EpiMT.

We sought to characterize the spatiotemporal emergence of AV EPDCs. However, many factors expressed within this population, including *Postn*, *Sox9*, and *Nfib*, are also activated during EndMT. *Hic1*, a transcriptional repressor, presented itself as a factor specific to EpiMT[62]. To characterize the spatiotemporal emergence of these cells, we combined LacZ staining on *Hic1*[nLacZ/+] heart cryosections with

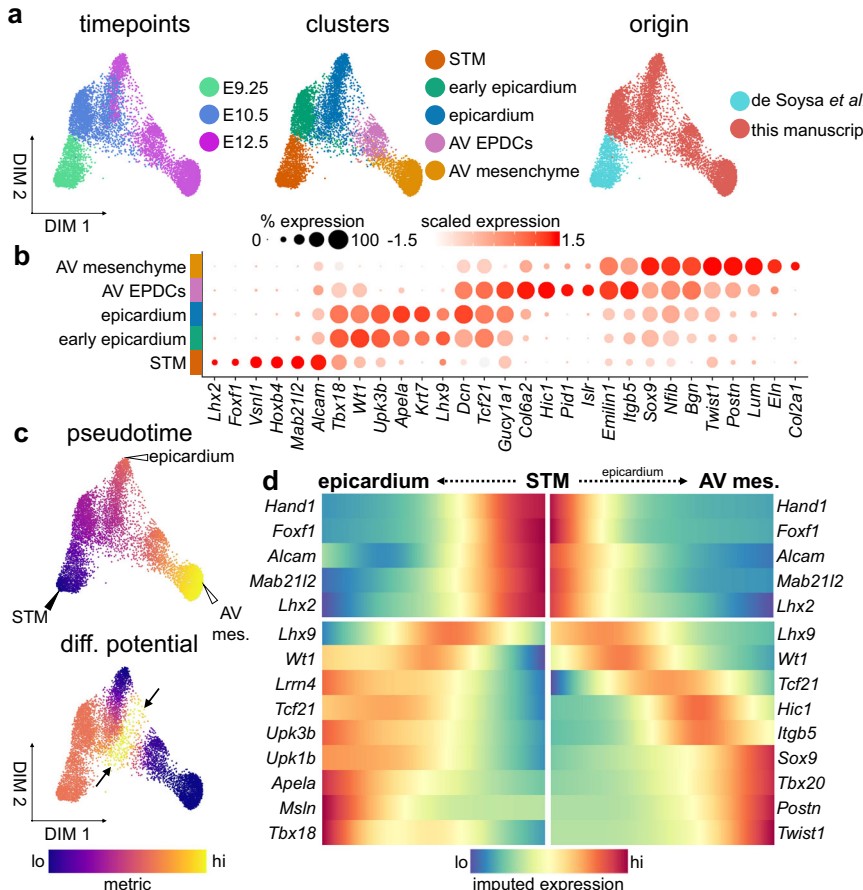

**Fig. 5 | The epicardial-to-mesenchymal transition progresses through a distinct cell type. a** Force-directed layouts of 4633 epicardial cells displaying timepoint, annotated cluster, and origin. **b** A dot plot displaying select genes enriched within epicardial clusters, including AV epicardial-derived cells (EPDCs) and mesenchyme. Size of the dot represents proportion of the cluster that expresses each gene. Colour indicates level of expression. **c** Force-directed layouts of 4633 epicardial cells displaying pseudotime and differentiation (diff.) potential. Arrow indicates differentiation potential maximum where AV EPDCs are specified. **d** Gene trends along pseudotime show bifurcation between epicardial and AV mesenchymal trajectories during the epicardial-to-mesenchymal transition. AV atrioventricular, epi epicardium, mes mesenchyme, STM septum transversum mesenchyme. See also Supplementary Fig. 5 and Supplementary Data 2.

fluorescence immunostaining[63,64]. We find that $Hic1^{nLacZ}$-expressing cells are present along the outer edge of the AVC, within the AV sulcus where EpiMT occurs at E12.5 and E14.5 (Fig. 6c, d). We find that some of these cells co-express SOX9 (arrowheads, Fig. 6c, d), indicating induction of AV mesenchymal identity, and we do not observe co-expression with endothelial ERG or myocardial TNNT2. We further observe that while WT1^hi $Hic1^{nLacZ\ lo}$ epicardial cells exist in a single-layered epithelium at E12.5 and E14.5 (yellow arrowheads, Supplementary Fig. 6c, d), $Hic1^{nLacZ\ hi}$ and WT1^lo AV EPDCs stratify and appear to adopt a more mesenchymal phenotype as they transdifferentiate, in line with our gene profile analysis. These findings reveal that a distinct AV EPDC population, expressing *Hic1*, emerges during EpiMT in the AV sulcus.

## Cell signalling microenvironment during EndMT and EpiMT

Identifying factors specific in driving EndMT and EpiMT initiation and progression is of interest, especially as the presence of endothelial and epicardial lineages in the AVC confounds interpretations of cell signalling on each of these processes individually[65]. We applied NicheNet to identify candidate factors that may induce and promote EndMT and EpiMT[66]. NicheNet shortlists candidate ligands and receptors, expressed by the defined sender and receiver cells, respectively, and ranks potential interactions along with downstream target genes using publicly available interaction databases.

We assessed which factors expressed by mesenchymal, epicardial, myocardial, and endocardial populations might signal towards endocardial cells to drive EndMT at E10.5 (Supplementary Fig. 7a and Supplementary Data 5). In our analysis, we see the activation of signalling pathways classically associated with EndMT initiation and mesenchymal development, including TGFβ superfamily[9,10,67], WNT[18], VEGF[68,69], and FGF[70] signalling. Interestingly, TGFβ1 is not predicted to influence EndMT; however, a variety of BMPs along with INHBA and GDF6 are identified as active drivers of mesenchymal identity (Fig. 7a). Cell-matrix interactions are prominent during EndMT, including endocardial receptors SDC1, DAG1, PTPRS, ITGA1, and ITGB1 interacting with NPNT, TNC, HSPG2, and type II and IV collagens (Fig. 7a). Mechanical signalling may play a role in the phenotypes observed in the hyaluronan[71], FN1[57], and POSTN[72] knockouts, for which the latter is already known to mediate atrioventricular development through integrin signalling. These findings suggest a multitude of ECM components provide cues to induce transdifferentiation through mechanical signalling[73]. Interactions between junctional adhesion molecules (JAMs), including F11R, JAM2, and JAM3, are identified, and while their role during AV development is unknown, they are essential drivers of EMT in the context of cancer metastasis, modulating morphology, migration, and cell polarity[74].

Downstream target prediction of ligands identified in our analysis supports the role of TGFβ superfamily members in driving EndMT.

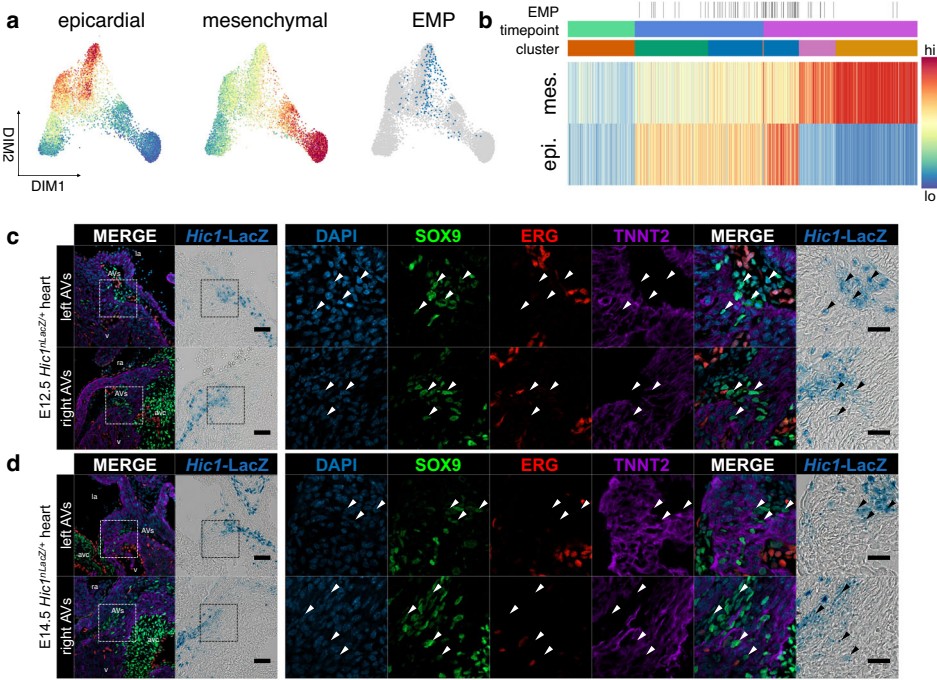

**Fig. 6 | Epithelial-mesenchymal plasticity during EpiMT. a** Force-directed layouts and (**b**) a heatmap of 4633 epicardial cells displaying average epicardial and mesenchymal gene profiles. Overlap in gene profile expression suggests emergence of EMP during EpiMT. Immunostaining of *Hic1^nLacZ/+* AV canals for SOX9, ERG, and TNNT2 at (**c**) E12.5 (*n* = 9) and (**d**) E14.5 (*n* = 6) shows co-expression of mesenchymal SOX9 and AV EPDC *Hic1^nLacZ* within AV sulcus, suggesting transition to mesenchyme (arrowheads). Scale bars: 100 μm, 25 μm in ROIs. AV atrioventricular, AVs AV sulcus, epi epicardium, mes mesenchyme. See also Supplementary Fig. 6 and Supplementary Data 2 and 4.

BMP2, BMP4, BMP5, BMP7, INHBA, and GDF6 are all predicted to promote the expression of transcriptional regulators of mesenchymal identity along with the expression of ECM components associated with AV cushion development (Supplementary Fig. 7b). BMP2 is predicted to be especially influential, potentially inducing the expression of a variety of critical mesenchymal genes, including *Sox9*, *Snai1*, *Msx1*, *Postn*, *Tagln*, and *Col1a1*. This is in line with the *Bmp2^fl/fl*;*Nkx2-5-cre* phenotype, where AV cushions and mesenchyme are absent due to the failure to initiate EndMT[10,58]. Of note, myocardial cues are predicted as the main drivers of EndMT (Supplementary Fig. 7b), which agrees with previous analyses that assessed AV identity induction[7,58].

We next assessed which factors might signal towards epicardial cells to drive AV EpiMT at E12.5 using NicheNet (Supplementary Fig. 7c and Supplementary Data 5). As seen in EndMT, we predict prevalent TGFβ superfamily signalling in the promotion of EpiMT, including various BMPs, GDF11, and INHBA. However, we further predict active signalling through TGFβ1 and TGFβ3, which are not expected to influence EndMT (Fig. 7c). PDGFA and FGF9 are predicted to signal via PDGFRα/β and FGFR1/2, respectively. Mechanical signalling via integrins and syndecans is also featured prominently, suggesting a role for cell-matrix interactions in promoting EpiMT (Fig. 7c). Downstream target prediction of ligands identified in our analysis supports the role of TGFβ superfamily signalling in driving EpiMT and promoting a mesenchymal cell fate. Our analysis suggests that TGFβ1 is the most influential factor in promoting EpiMT and may activate a variety of regulators associated with AV EPDC or AV mesenchymal identity, including *Hic1*, *Sox9*, *Snai2*, *Postn*, *Tagln*, integrins, and type III and V collagens (Supplementary Fig. 7d). Other TGFβ family members are predicted to act upstream of genes associated with cell proliferation, like *Junb* and *Myc*, along with genes associated with a mesenchymal phenotype, like *Acta2*, *Zeb2*, and *Mecom*. Lastly, we assessed which factors may promote the proliferative pulse in AV mesenchymal cells at E12.5 (Supplementary Fig. 7e–g and Supplementary Data 5). We identify candidate signals from the endocardium, like TGFβ1 and

WNT ligands, which are known mediators of valve mesenchymal proliferation[46,75]. We further identify candidate factors, including MDK, FGF9, and IGF1, among others, which may drive this proliferative pulse (Supplementary Fig. 7f, g).

## Discussion

These data provide an in-depth look into how the early AV valves develop through the computational deconvolution of EndMT and EpiMT, revealing the molecular details underlying these processes. We define previously uncharacterized cellular heterogeneity within the AVC and propose mechanisms of lineage emergence. This includes distinct AV endocardium, VEC, and mesenchymal populations, along with a population of AV EPDCs.

Our data suggest that fluid shear stress induced by blood flow contributes to the development of EMP and endocardial cell stratification during EndMT. The early endocardium requires the maintenance of tight junctions to sustain endothelial barrier and cardiac functions. We propose that endocardial stratification during EndMT through the adoption of EMP allows for the retention of its barrier function, maintaining tight endothelial cell-cell contacts within the apical most cells in this region during the formation of the AV mesenchyme. The adoption of EMP during EndMT is closely associated with the activation of mechanoresponsive genes and high differentiation potential, indicating a role for hemodynamic forces in regulating cell fate identity. We further observe the developmental arrest of endocardial cells that exhibit EMP within the AVC when *Sox9* is conditionally ablated along with mesenchymal gene dysregulation and transdifferentiation to myofibroblasts. This suggests that at least two distinct processes function to promote AV EndMT. The first is SOX9-independent, involving the adoption of EMP possibly in part due to fluid shear stress, while the second is SOX9-dependent and involves the complete transdifferentiation to AV mesenchyme. Differences in gene expression in the *Sox9* cKO suggest that cues involving BMPs, FGFs, PDGFs or integrin-FN1 interactions could be

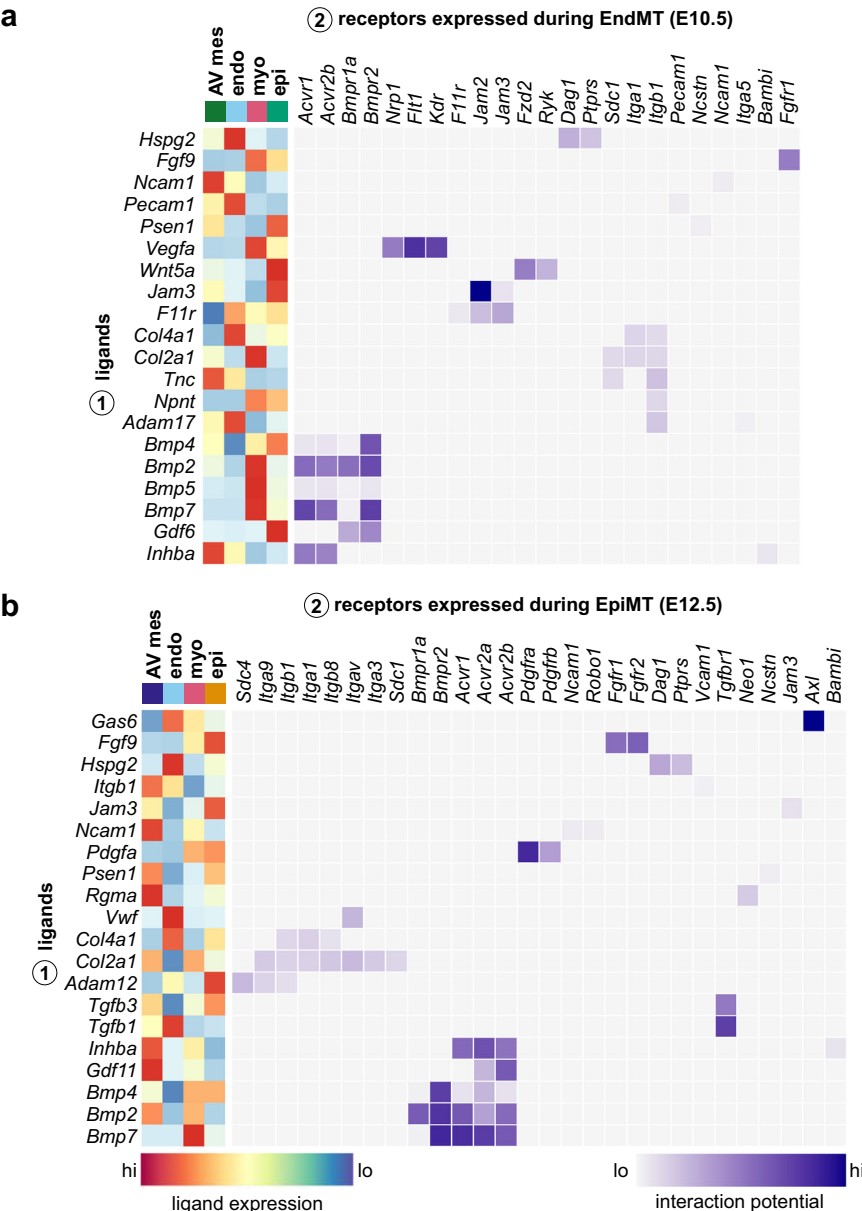

**Fig. 7 | Deconvolution of signalling microenvironment during EndMT and EpiMT. a** A heatmap displaying select candidate ligand-receptor interactions signalling toward endocardial cells at E10.5 during EndMT, including scaled and log-normalized expression of ligands from sender populations. **b** A heatmap displaying select candidate ligand-receptor interactions signalling toward epicardial cells at E12.5 during EpiMT, including scaled and log-normalized expression of ligands from sender populations. AV atrioventricular, endo endocardium, epi epicardium, myo myocardium. See also Supplementary Fig. 7 and Supplementary Data 2 and 8.

responsible. The latter is supported by the reduced expression of integrins essential for FN1 adhesion in the *Sox9* cKO, along with the accumulation of stratified endocardium within the *Fn1* KO, which echoes the phenotype we observe in the *Sox9* cKO[57]. Overall, this suggests a role for integrin-ECM interactions in promoting an AV mesenchymal cell fate.

Our analyses further provide insight into AV EpiMT, where we reveal the development of AV EPDCs as early as E10.5. AV EPDCs express a variety of factors associated with EMT, along with some unique genes, like *Hic1*, a tumour suppressor[62] and biomarker of adult mesenchymal progenitors[63,76]. Staining of *Hic1*[nLacZ/+] hearts shows the presence of these cells within the AV sulcus by E12.5. These cells maintain low WT1 and activate SOX9 as they transdifferentiate towards an AV mesenchymal cell state. Our cell signalling analyses suggest TGFβ superfamily, PDGF, FGF, and mechanical signalling may play a role in the activation of this cell type.

Finally, our transcriptional data indicate that EndMT and EpiMT share many features despite their distinct cellular origins. Expression of canonical EMT transcription factors, including *Snai1/2, Zeb1/2*, and *Twist1*, is observed in both contexts[77]. EndMT and EpiMT are both initiated within an epithelium and exist transiently in a stratified state, where the induction of EMP is associated with high cell lineage plasticity. Future research to validate the presence of EMP and potential for differentiation will likely employ dual lineage-tracing techniques, incorporating epithelial and mesenchymal drivers along with an intersectional reporter, to trace the fate of cells exhibiting EMP during valve development[78]. Further, the absence of endocardial and epicardial transcriptional legacies in AV mesenchyme, strongly suggest that mesenchyme generated during EndMT and EpiMT converge to an indistinguishable cell identity, though validation would likely require dual lineage tracing combined with cell sorting or scATAC-seq. However, differences also exist between these processes. This includes the

presence of hemodynamic forces in promoting EndMT, differences in the cell signalling microenvironments, and process-specific expression of transcription factors, including *Nfatc1* and *Hic1* in EndMT and EpiMT, respectively.

In summary, these data advance our understanding of the initiation and progression of AV EMTs, providing a detailed assessment of these processes at single-cell resolution. These data will inform researchers studying cardiogenesis, congenital heart defects, in vitro models of EndMT, like ES cell differentiation of cardiac valve lineages[79], as well as the recapitulation of developmental processes in disease and regenerative contexts. These data may be of special interest to cancer researchers studying EMT, during which the adoption of EMP is generally associated with more aggressive disease[33].

## Methods

All mice were maintained in accordance with the University of British Columbia's Animal Care Committee's standards under protocol numbers A20-0281, A20-0282 (principal investigator, P.A.H.), and A19-0221 (principal investigator T.M.U.) in pathogen-free conditions.

### Reagents and Resources

Detailed information related to reagents and resources used in this study are available in Supplementary Data 6.

### Mouse husbandry

Elite CD-1 (Charles River), *Hic1^(nLacZ/+)* [63], *tdTomato* (*Gt(ROSA) 26Sor^(tm9(CAG-tdTomato)Hze)*)[80], *Sox9^fl* (*B6.129S7-Sox9^(tm2Crm)/J*)[56], *Tie2-cre* (*Tg(Tek-cre)1Ywa*)[81] mice were used. Homozygous *tdTomato* females and hemizygous *Tie2-cre* males were bred for endothelial lineage tracing analyses, including generation of E12.5 AVC scRNA-seq libraries. Homozygous *Sox9^(fl/fl)* females were bred with *Sox9^(fl/fl);Tie2-cre* males to conditionally ablate *Sox9* in endothelial lineages, and were used for immunostaining and generation of E10.5 *Sox9* cKO AVC scRNA-seq library. *Hic1^(nLacZ/+)* embryos were used for LacZ staining and co-immunostaining of AV EPDCs. CD-1 mice were used for all other experiments. Up to five mice were housed per cage and were maintained on a regular chow diet *ad libitum* on a 12-h light-dark cycle. Embryos from timed matings were considered E0.5 at noon of the day a vaginal plug was observed. Embryos were collected at experimentally determined timepoints in ice-cold phosphate-buffered saline (PBS) under a Leica MZ6 dissecting scope. Embryo sex was not determined.

Primer sequences for genotyping are listed in Supplementary Data 7. *Tie2-cre;tdTomato* embryos were genotyped using a Zeiss AxioZoom V16 to observe *tdTomato* fluorescence. The presence of *Hic1^(nLacZ)* transgene in embryos was done through overnight incubation of tissues in LacZ staining solution (2 mM MgCl2, 0.01% deoxycholate, 0.02% NP-40, 5 mM potassium-ferricyanide, 5 mM potassium-ferrocyanide, and 1 mg/ml X-gal in PBS) at 37 °C and visualization with a Leica MZ6 dissecting scope. Sex-specific differences were not anticipated and embryo sex was not determined.

### Single-cell dissociations of atrioventricular canals

For dissociations, siliconized tips were used for all steps, and centrifugations were performed at minimum acceleration. For E10.5 AVC replicate 1 and FACS analyses related to Supplementary Fig. 4, we used a dispase/collagenase dissociation protocol, where AVCs were collected using fine dissecting forceps, ensuring the removal of the outflow tract, atria, and ventricles. The AVC, including AV cushions as well as overlying myocardium and epicardium were retained. AVCs were washed in PBS and incubated for 5 min at 37 °C in 1 ml PDEF (10 μg/ml DNAse I, 4 mM EDTA, and 2% foetal bovine serum [FBS] in PBS) with dispase/collagenase (2.5μU/ml dispase, 0.0025% collagenase type I) to dissociate tissues into single-cells. Tissues were gently triturated using a 1000 μl pipette before and after enzymatic digestion to facilitate

dissociation. Single-cell suspensions were filtered through 40 μm cell strainers into 5 ml FACS tubes to remove debris and incompletely dissociated cells. The strainers were washed with PDEF to collect the remaining cells. The filtrate was subsequently centrifuged at 1000 g for 4 min to sediment single cells and the supernatant was removed.

We found that while dissociation with dispase/collagenase successfully captured endothelial- and epicardial-derived cells, myocardial cells are absent. Therefore, for all other libraries at E10.5 and E12.5, single-cell dissociations were performed with TrypLE Express as was done in[36]. *Tie2-cre;tdTomato* embryos were used for generation of E12.5 scRNA-seq libraries. Littermate *Sox9^(fl/fl);Tie2-cre* and *Sox9^+;Tie2-cre* embryos were used for generation of E10.5 *Sox9* cKO and control scRNA-seq libraries (Supplementary Fig. 1a). Tissues were collected from E10.5 and E12.5 embryonic hearts, ensuring removal of cardiac chambers and outflow tract. AVCs were washed for 5 min in PBS to remove red blood cells. PBS was removed and replaced with 250-1000 μl TrypLE Express. Tissues were gently triturated, then incubated at 37 °C for 5–15 min until the suspension was homogeneous and pieces of tissue were not visible. Cell suspensions were gently triturated part way through for longer incubations and again at the end. Single-cell suspensions were filtered through 40 μm cell strainers into 5 ml FACS tubes to remove debris and incompletely dissociated cells. The strainers were washed with PDEF to collect the remaining cells. The filtrate was subsequently centrifuged at 1000 g for 4 min to sediment single cells and the supernatant was removed. For E10.5 *Sox9* cKO and littermate control libraries, AVCs were collected and left on ice for 90 min while genotyping was performed. *Sox9^(fl/fl);Tie2-cre* cKO AVCs or *Sox9^(+/+)* and *Sox9^(fl/+);Tie2-cre* AVCs were then pooled and dissociated.

### Fluorescence-activated cell sorting

For single-cell library preparation, sedimented single-cells from embryonic AVCs were resuspended in PDEF and sorted using a FACSARIA III cell sorter (BD Biosciences) with a 130 μm nozzle in purity mode. All cell suspensions were stained with 1:5000 300 μM DAPI to purify live cells. Cellular debris and cell doublets were excluded based on forward and side scatter. To remove hematopoietic populations, filtered, dissociated AVC suspensions were resuspended in 2% FBS in PBS, and stained with APC-conjugated anti-Ter119 (1:200) at 4 °C for 15 min. After staining, cell suspensions were washed three times in PDEF, centrifuging at 1000 g for 4 min, and were ultimately resuspended in PDEF. APC+ fractions were excluded from scRNA-seq library construction (Supplementary Fig. 1a).

For FACS analyses related to Supplementary Fig. 4, sedimented single-cells from individual embryonic AVCs were resuspended in 2% FBS in PBS and incubated with goat anti-PDGFRα (1:100), FITC-conjugated rat anti-CD31, and APC-conjugated rat anti-Ter119 for 15 min at 4 °C. Cells were centrifuged at 1000 g for 4 min and re-suspended in 2% FBS in PBS and stained with donkey anti-goat Alexa 568 for 15 min at 4 °C. Cells were centrifuged at 1000 g for 4 min and re-suspended in PDEF with 1:5000 300 μM DAPI to identify dead cells. Cells were sorted using a FACSARIA III cell sorter (BD Biosciences) with a 130 μm nozzle in purity mode. Cellular debris and cell doublets were excluded based on forward and side scatter. CD31+ PDGFRα+ cells were collected for single-cell histology.

### Single-cell RNA-seq library generation

Sorted single-cell suspensions were counted and diluted to a final concentration in low EDTA PDEF (0.1 mM EDTA) and cellular suspensions were loaded on a Chromium Controller to generate single-cell gel bead emulsions, targeting 7650–10,000 cells depending on the tissue and embryonic stage (Supplementary Data 1). For AVC scRNA-seq, three E10.5 libraries from WT and *Sox9* cKO hearts and two libraries from E12.5 *Tie2-cre;tdTomato* AVCs were created. Single-cell 3′ RNA-seq libraries were generated according to the manufacturer's instructions (Chromium Single Cell 3′ Reagent v2 Chemistry Kit, 10X Genomics,

Inc.). Libraries were sequenced to a depth of 50-175,000 reads per cell on an Illumina Nextseq system (E10.5 and E12.5 libraries). Refer to Supplementary Data 1 for library quality metrics.

### Read alignment

E7.75, E8.25, and E9.25 single-cell RNA-seq heart data from de Soysa et al. [36] was downloaded from GEO (accession number GSE126128). FASTQs from all libraries, including publicly available data, were aligned and quantified using the Cell Ranger Single Cell Software Suite version 6.0.0 (10X Genomics, Inc.) against the custom-built mm10 (GRCm38_GtC) reference genome containing *tdTomato*, *Cre*, and *Gfp* sequences. Cell Ranger was used to identify droplets containing cells for each library individually, identifying cells based on the number of UMIs and gene expression profiles. Aligned reads from E12.5 epicardium and EPDC single-cell RNA-seq data from Quijada et al. [27] was downloaded from GEO (accession number GSE154715).

### Single-cell data processing

All data processing and analyses were performed in R and Python using standard protocols from previously published packages. The number of cells used in each of the statistical analyses is indicated in the figures or figure legends. Further quality metrics were calculated using the calculateQCMetrics in the scater package in R[82], and cells were subsequently filtered based on unique features (>2000 genes) and mitochondrial RNA content (<20%).

Size-factor normalization, feature selection, dimensionality reduction, and doublet identification were performed using the scran and scater packages in R[82,83]. Briefly, filtered cells were loosely clustered using the quickCluster function, then size-factor normalized using the computeSumFactors function, taking both library sizes and quickCluster ID into account. Single-cell transcriptomes were subsequently log-transformed using the normalize function in scater to correct for biases between cells, considering size factors. Highly variable genes were selected based on gene expression levels and variance across all cells to highlight biological differences across datasets. Ribosomal, mitochondrial, Y-chromosome genes, and *Xist* were removed from highly variable genes. Doublet scores were calculated using the doubletCells function in scran, which assigns each cell a score that is defined as the ratio of the density of simulated doublets to the squared density of original cells in the neighbourhood of that cell[83].

PCA dimensionality reduction and mutual-nearest neighbours[84] or Harmony[34] batch correction was performed to correct for non-biological confounders between replicates and timepoints. PCA was done using the RunPCA function in Seurat[85]. For MNN batch correction, fastMNN from the scran package was employed, and the first 100 principal components were used as input. $k = 20$ neighbours were used for the calculation of mutual-nearest neighbours. Harmony uses timepoint metadata to construct augmented affinity matrices, which uses mutual-nearest neighbours across consecutive timepoints to correct for batch effects. Log-normalized expression-matrices, containing data for highly variable genes only, were used as inputs. A k of 20–35 was used for the mutual nearest neighbour calculation for the EndMT and EpiMT lineage analyses. The augmented affinity matrices were subsequently used to calculate the force-directed layouts.

Dimensionality reduction, like uniform manifold approximation and projection (UMAP) was performed using the runUMAP function in scater, respectively using the first 100 principal components as inputs. Force-directed layouts to display cell type diversity were calculated using adjusted affinity matrices generated with the Harmony package in Python[34]. Graph-based clustering and Wilcoxon rank-sum test for differential gene expression were performed using phonograph[39] and Seurat[85] packages in Python and R, respectively. For phenograph clustering, a k of 150 to 300 was used and the normalized expression matrix of highly variable genes served as the input. The FindAllMarkers function in Seurat was used to identify differentially expressed genes.

Globally enriched genes within a cluster were identified based on a log fold-change of 0.25 with an expression threshold of >25% within a population using two-sided Wilcoxon rank-sum test. Only positively enriched genes with a false-discovery rate (FDR) of 0.05 or lower are included. To better characterize the phenotypic identities of endocardial- and epicardial-derived lineages, we calculated an average gene profile for AV mesenchymal, endothelial, epicardial, or fluid shear stress[86] identities, using annotated lists of biomarkers and master regulators (Supplementary Data 4). Gene expression profiles were calculated for each single cell in the EndMT or EpiMT datasets by calculating the average log-normalized expression of annotated lists of biomarkers and master regulators of endothelial identity, epicardial identity, mesenchymal identity, or response to fluid shear stress (Supplementary Data 4). Classification of cells to gene profile categories (e.g. exhibiting EMP), was done through determining conservative minimum thresholds in each gene profile for meeting endothelial, epicardial, or mesenchymal identity (Supplementary Figs. 3a and 6a). EMP in generally defined as the strong co-expression of factors associated with mesenchymal and epithelial identity. Cells that met the threshold of both epithelial and mesenchymal profiles were considered to be exhibiting EMP. While EMP represents a continuum, we provided these data as a binary for easy visualization. Batch-associated genes were identified based on a log fold-change of 0.5 with an expression threshold of >75% within a batch using two-sided Wilcoxon rank-sum test. Gene ontology analysis for each batch-enriched list of genes was performed and, along with the gene lists, can be found in Supplementary Data 3[87]. Only positively enriched genes with a false-discovery rate (FDR) of 0.05 or lower are included. For visualization, gene profiles were mean-scaled for comparison and heatmaps were generated in R using pheatmap with clustering done using the ward.D2 method.

Palantir single-cell differentiation trajectories, which infer pseudotime progression, cell fate decisions, and terminal cell states from a defined 'start cell', were calculated using diffusion components from Harmony augmented affinity matrices discussed in the previous section[34,44]. 'Start cells' for each trajectory were selected based on current developmental paradigms, including an early endothelial progenitor for the endocardial lineages (EndMT), and a lateral plate mesoderm (LPM) progenitor for the epicardial lineages (EpiMT). 30 nearest-neighbours and 700 to 1000 waypoints were used as inputs for the generation of Palantir differentiation trajectories. For all trajectories calculated, the Pearson's correlation of Palantir results over a range of k nearest-neighbours demonstrates that pseudotime estimation, differentiation potential, and branch probabilities to termini are robust with respect to changes in the number of nearest-neighbours, k (Supplementary Figs. 2j and 5e). Gene expression was visualized after imputation with the MAGIC algorithm to denoise the single-cell data and fill in dropouts[88]. Gene expression trends along pseudotime were calculated based on branch probabilities and generalized additive models, showing the stepwise transcriptional progression of endodermal and endothelial lineages along pseudotime using Palantir[44].

### NicheNet cell signalling analysis

The transcriptional influence of ligands during EndMT and EpiMT was predicted by applying NicheNet[66]. NicheNet predicts which ligands produced by a sender cell are the most active in affecting gene expression in a receiver cell, through the correlation of ligand activity with genes previously characterized as targets of their downstream pathways.

To analyze the cell signalling microenvironment during AV valve development, we applied NicheNet to predict which ligands influence transcription during EndMT and EpiMT using cells from E10.5 and E12.5, respectively. At E10.5, early endocardial and VEC II clusters were defined as receiver cells, and all populations, including endocardium, mesenchyme, epicardium, and myocardium, were used as sender

populations (Supplementary Fig. 7a). At E12.5, epicardial clusters were defined as receiver cells, and all populations, including endocardium, mesenchyme, epicardium, and myocardium, were used as sender populations (Supplementary Fig. 7c). Gene sets of interest were defined based on differential gene expression analysis where the top differentially-enriched genes based on false-discovery rate were used (average log2 fold change >0.25; expressed in at least 25% of the population of interest). This includes genes specifically enriched within the mesenchyme over endocardium at E10.5 and AV EPDCs over typical epicardium at E12.5 for EndMT and EpiMT analyses, respectively (Supplementary Data 8). A background gene set includes all other genes that are highly expressed, but not differentially expressed between these populations. NicheNet's ligand-target model was converted from human to mouse genes using the convert_human_to_mouse_symbols function. The complete results from the NicheNet analysis can be found in Supplementary Data 5.

The circlize package in R was used to separately represent the regulatory networks between the predicted ligands and their targets expressed in receiver cell types[89]. Ligands were grouped and annotated based on the cell of origin. The width and opacity of links were determined based on the ligand-receptor interaction weights and ligands' activity scores, respectively.

## Tissue processing for histology

Embryonic tissues for fluorescence immunohistochemistry were collected from timed matings at experimentally determined timepoints. Tissues were washed in PBS to remove blood. The heads of whole embryos and ventricles of dissected hearts were carefully punctured with dissecting forceps to allow for better washing. Tissues were incubated overnight in 4% paraformaldehyde (PFA) at 4 °C, washed with PBS, and dehydrated through a methanol gradient (25%–50%–75%–100% methanol in PBS, room temperature, 5 min each).

## Immunostaining of tissue cryosections

Dehydrated tissues were rehydrated to PBS from methanol (75%–50%–25% methanol in PBS, room temperature, 5 min each), then washed in PBS for 5 min. Tissues were then washed through a sucrose gradient (15%–30%–60% sucrose in PBS, 4 °C, 1–12 h each), and overnight in 1:1 60% sucrose:optimal cutting temperature compound (OCT, TissueTek) at 4 °C. Cryoprotected tissues were embedded in OCT, flash-frozen, and stored at −80 °C.

8 μm sections were collected using a Leica CM3050S cryostat at −25 °C with SuperFrost Plus slides. Sections were circumscribed with an Elite PAP pen (Diagnostic Biosystems K039) and slides were subsequently placed in an opaque humidity chamber. Sections were refixed for 10 min at room temperature in 4% PFA in PBS, washed three times with PBS for 5 min, then incubated in block solution (5% bovine serum albumin, and 0.01% Triton-X in PBS) for 1 h at room temperature. Sections were subsequently incubated with primary antibodies in block solution overnight at 4 °C. These include anti-CD31 (1:50), anti-ERG (1:50), anti-GATA4 (1:50), anti-MCM-2 (1:100), anti-NFATc1 (1:15), anti-PCNA (1:100), anti-PDGFRα (1:50), anti-pH3 (1:100), anti-SOX9 (goat−1:50; rabbit−1:500), anti-TNNT2 (1:15), anti-VIM (1:50), and anti-WT1 (1:50). The next day, sections were washed with PBS three times for 5 min and incubated with species-specific Alexa Fluor 488-, 568-, 594-, or 647-conjugated secondary antibodies in block solution (all 1:500) for 1 h at room temperature. After secondary incubation, sections were washed with PBS three times for 5 min, incubated with 1:1000 300 μM DAPI to counterstain nuclei, and then washed again with PBS three times for 5 min. Slides were mounted using a minimal volume (-100 μl) 25 mg/ml DABCO in 9:1 glycerol:PBS. Coverslips were fixed in place using clear nail polish.

Images were captured using a Nikon Instruments Eclipse Ti confocal laser microscope. Image capture, processing, and quantification was done using NIS Elements 5.0 and Fiji in ImageJ[90], with brightness, contrast, pseudo-colouring adjustments, and z-stack alignments applied equally across all images in a given series.

## Immunostaining of FACS-sorted single-cells

Sorted cells were sedimented at 1000 g for 4 min, re-suspended in 50 μl of PDEF, and deposited onto SuperFrost Plus slides. Cells were left to settle for 30 min and then fixed with 4% PFA in PBS. Slides were placed in a humidity chamber for staining. Blocking, antibody incubation, and mounting were done as described for tissue cryosections.

## Combination LacZ and immunostaining of Hic1[nLacZ/+] cryosections

*Hic1[nLacZ/+]* tissue cryopreservation and LacZ staining were done as outlined in Scott & Underhill[64]. Fixed *Hic1[nLacZ/+]* hearts were washed through a sucrose gradient (15%–30%–60% sucrose in PBS, 4 °C, 1–12 h each), and overnight in 1:1 60% sucrose:OCT 4 °C. Cryoprotected tissues were embedded in OCT (TissueTek), flash-frozen, and stored at −80 °C. 8 μm sections were collected using a Leica CM3050S cryostat at −25 °C with SuperFrost Plus slides. Sections were circumscribed with an Elite PAP pen (Diagnostic Biosystems K039) and slides were subsequently placed in an opaque humidity chamber. Sections were washed with PBS three times for 10 min to remove OCT compound and incubated at 37 °C overnight in LacZ staining solution (2 mM MgCl2, 0.01% deoxycholate, 0.02% NP-40, 5 mM potassium-ferricyanide, 5 mM potassium-ferrocyanide, and 1 mg/ml X-gal in PBS). The next day, slides were washed in distilled $H_2O$ and washed three times for 30 min with PBS. Autofluorescence was quenched for 45 min with sodium borohydride (10 mg/ml in PBS), washed again with PBS, and then incubated in block solution (5% bovine serum albumin, and 0.01% Triton-X in PBS) for 1 h at room temperature. Sections were subsequently incubated with primary antibodies in block solution overnight at 4 °C. These include anti-CD31 (1:50), anti-ERG (1:50), anti-SOX9 (goat −1:50), anti-TNNT2 (1:15), and anti-WT1 (1:50). The next day, sections were washed with PBS three times for 5 min and incubated with species-specific Alexa Fluor 488-, 568-, 594-, or 647-conjugated secondary antibodies in block solution (all 1:500) for 1 h at room temperature. After secondary incubation, sections were washed with PBS three times for 5 min, incubated with 1:1000 300 μM DAPI to counterstain nuclei, and then washed again with PBS three times for 5 min. Slides were mounted using a minimal volume (-100 μl) 25 mg/ml DABCO in 9:1 glycerol:PBS. Coverslips were fixed in place using clear nail polish.

Images were captured using a Nikon Instruments Eclipse Ti confocal laser microscope, capturing transmitted and reflected light. Image capture or processing was done using NIS Elements 5.0 and Fiji in ImageJ[90], with brightness, contrast, pseudo-colouring adjustments, and z-stack alignments applied equally across all images in a given series. To image stained *Hic1[nLacZ/+]* tissue, transmitted light was captured using the blue, green, and red lasers, merged to form an RGB image, then shade-corrected and white balanced using ImageJ.

## Statistics and reproducibility

No statistical method was used to predetermine sample size. The experiments were not randomized and no data was excluded from analysis. Researchers were not blinded during data acquisition and analysis. All statistical analyses were performed and graphs generated using R and Python. scRNA-seq analyses were performed as described above. The number of cells used in each of the statistical analyses is indicated in the figures or figure legends.

For single-cell RNA-seq data processing, the Cell Ranger Single Cell Software Suite version 5.0 (10X Genomics, Inc.) was used for alignment, de-multiplexing, and UMI counting using the default parameters. Sequencing metrics are presented in Supplementary Data 1. From 14 samples, 53,521 single cells were analyzed, ranging

from 1313 to 11,166 cells collected per sample. Median reads per cell ranged from 46,691 to 175,383 with median UMI of 14,012 to 36,267 per cell. Cells with <1000 expressed genes or with mitochondrial gene expression accounting for >20% of total RNA content were considered of low quality and excluded from downstream analyses.

After single-cell data normalization, clustering, and dimensionality reduction, marker genes for each cluster were identified using the FindAllMarkers or FindMarkers functions in Seurat using two-sided unpaired Wilcoxon rank-sum test. Genes with an FDR < 0.05 were considered statistically enriched within a cluster.

Pearson's correlation coefficients of a ligand's target gene predictions and the observed transcriptional response in the target population was used to define ligand activity for NicheNet ligand-target gene analysis. The top quartile of scores of interactions were visualized and the entire interactome for both EndMT and EpiMT can be visualized in Supplementary Data 4.

Pearson's correlation of Palantir results over a range of k nearest-neighbours demonstrates that pseudotime estimation, differentiation potential, and branch probabilities to terminal states are robust with respect to changes in the number of nearest-neighbours, k.

### Reporting summary
Further information on research design is available in the Nature Portfolio Reporting Summary linked to this article.

## Data availability
The single-cell RNA-sequencing data generated during this study have been deposited in the ArrayExpress database at EMBL-EBI (www.ebi.ac.uk/arrayexpress) under the accession number E-MTAB-11795. The processed data can also be explored at the Broad Single-cell Portal under the accession number SCP1897. Source data are provided with this paper.

## Code availability
Only publicly available tools were used in data analysis as described wherever relevant in the Methods.

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

## Acknowledgements

We thank F. Lynn, F. Rossi, and T. Stephan for constructive discussions as well as A. Shanks and P. Stirling for antibodies. This work was supported by the Canadian Institutes of Health Research (PJT-159512 to P.A.H.; PJT-149026 to T.M.U.), the Canadian Foundation for Innovation, and the British Columbia Knowledge Development Fund. This research was enabled in part by the Digital Research Alliance of Canada (www.alliancecan.ca). J.L. is the recipient of an NSERC PGSD scholarship and CIHR Postdoctoral Fellowship. S.D. is the recipient of a CIHR CGSM scholarship.

## Author contributions

J.L., R.C., V.C.G., and P.A.H. conceived the research. J.L., R.C., V.C.G., B.M.F., and A.T. performed preliminary analyses. J.L., R.C., and M.A. collected and dissected embryos. J.L. performed cell dissociations, FACS experiments, and histology. J.L. and M.C.-R. quantified histological data. J.L. and S.D. performed computational analysis of sequenced data. J.L and P.A.H. analyzed and interpreted the data, and wrote the manuscript with input from all authors. T.M.U. and P.A.H. performed supervisory roles and acquired funding.

## Competing interests

The authors declare no competing interests.
