## [Peer Review File · Nature Communications]

Cell diversity and plasticity during atrioventricular heart valve EMTsEditorial Note: This manuscript has been previously reviewed at another journal that is not operating a transparent peer review scheme. This document only contains reviewer comments and rebuttal letters for versions considered at *Nature Communications*.

REVIEWER COMMENTS

Reviewer #1 (Remarks to the Author):

My comments have been addressed and the revised manuscript is improved. I have no additional comments.

Reviewer #3 (Remarks to the Author):

Review of revision of NCOMMS-23-06809A, Lotto et al. "Single-cell transcriptomics reveals diversity during heart valve epithelial-to-mesenchymal transitions."

In our review of the original manuscript, we found that the study is a nice exploration of EndMT and EpiMT and potentially advances our knowledge of both processes. However, we identified several areas in need of improvement. In the revised manuscript, some of the areas needing improvement were addressed, but in most cases the authors were not responsive to the comments.

Importantly, the contribution of EndoMT and EpiMT to heart valves has been analyzed in the same embryo using orthogonal lineage tracing systems (PMID: 30111655). This manuscript shows that at E12.5, the timepoint analyzed in this study, EpiMT contributes to a small (~5%) of valve mesenchyme cells in the mural leaflets. The fraction of valve mesenchyme that arises from epicardium increases over time, reaching ~50% at E17.5 and >80% at P7. These data are consistent with the images from an independent lineage tracing study (PMID: 22546693). These data suggest that this study has not obtained the appropriate population of valve mesenchyme cells to use as the terminus for EpiMT.

Major points.

1. Histological data continues to lack quantitative analysis.
2. The authors combined their data with published data (De Soya et al). They perform integrative analysis but do not show that their analysis overcomes batch effects. In their reply, the authors indicate that the method used, Harmony, is not designed to overlay datasets completely but rather to stitch together datasets in a time series. There are at least two variables that could account for differences between cells in the different data sets: (1) time point; (2) batch effect. The data sets do not appear to intermingle well (Fig. S1G) and it is not possible to determine if this is due to a difference in the time point or a technical difference due to batch effect. Regardless of the analysis method used, the authors need to provide data that resolves this question. As mentioned in the initial review, there are many eligible data sets that could be included in the analysis. The authors do not provide a solid reason for selectively using the De Soya et al data set. Finding similar results in multiple data sets would go far to strengthening the conclusion.
3. The authors also only obtained a single replicate of the Sox9 cKO samples so reproducibility was not demonstrated. The authors replied that the one library was composed of multiple embryos from two litters and that other manuscripts are published with data sets

that do not show reproducibility. Multiple embryos in one library means the library represents the average of multiple embryos but does not demonstrate reproducibility. Low scientific rigor in other manuscripts is not good justification for low rigor.

4. In the initial review we requested improved clarity on how differentiation potential and EMP were defined and calculated. Although the authors indicated in their reply that the information is included in the revised methods, these terms remain opaque. Moreover, there is no validation that these scores are meaningful. In some cases, EMP is indicated as a binary value, implying use of a threshold. How the threshold was established and validated is also not clearly stated. Does this threshold differ between EpiMT and EndMT?

5. The framework of the epicardial trajectory analysis remains confusing. Consistent with literature, the authors state “AV EPDCs are specified from epicardium, and not directly from a mesodermal progenitor earlier in development.” If this is the case, then why is the trajectory analysis set up so that AV EPDCs are alternate termini from an STM progenitor start point, rather than a sequence of STM progenitor -> epicardium -> AV EPDC?

6. The authors specified AV mesenchyme as a terminus of epicardial differentiation. It seems this is the same AV mesenchyme used for analysis of EndoMT -- making the assumption that both EndoMT and EpiMT yield similar valve mesenchyme cells, as opposed to distinct subsets of valve mesenchyme cells. What is the basis for this assumption? As described in PMID 22546693 and 30111655, EpiMT contributes to only a small fraction (~5%) of AV valve mesenchyme at this stage.

7. In the revised manuscript, the authors claim identification of a “unique AV EPDC population”. The label “unique” is used multiple times to describe this population. What is the meaning of “unique” – unique from what? The authors identify some markers that distinguish chamber and AV EPDCs (Fig. S5H-I), but it appears the main marker used, Hic1-nLacZ, does not distinguish these types of EPDCs (Fig. 5C).

8. Since the authors declined to validate the cell signaling analysis, particularly the prediction that AV mesenchyme and endocardium ligands contribute most to EpiMT, this should be moved to the discussion and stated with the caveat that it has not been experimentally validated.

Cell diversity and plasticity during atrioventricular heart valve EMTs

NCOMMS-23-06809-T

We would like to thank the reviewers for their constructive comments. We have edited and revised the manuscript to address critiques including those related to batch effects, incorporation of publicly available data, and discrimination of AV mesenchyme derived from EndMT and EpiMT.

We have included point-by-point responses in red to the reviewer comments below.

Reviewers' comments:

Reviewer #1 (Remarks to the Author):

My comments have been addressed and the revised manuscript is improved. I have no additional comments.

We thank reviewer #1 for their critiques, which have greatly improved this manuscript.

Reviewer #3 (Remarks to the Author):

Review of revision of NCOMMS-23-06809A, Lotto et al. "Single-cell transcriptomics reveals diversity during heart valve epithelial-to-mesenchymal transitions."

In our review of the original manuscript, we found that the study is a nice exploration of EndMT and EpiMT and potentially advances our knowledge of both processes. However, we identified several areas in need of improvement. In the revised manuscript, some of the areas needing improvement were addressed, but in most cases the authors were not responsive to the comments.

Importantly, the contribution of EndoMT and EpiMT to heart valves has been analyzed in the same embryo using orthogonal lineage tracing systems (PMID: 30111655). This manuscript shows that at E12.5, the timepoint analyzed in this study, EpiMT contributes to a small (~5%) of valve mesenchyme cells in the mural leaflets. The fraction of valve mesenchyme that arises from epicardium increases over time, reaching ~50% at E17.5 and >80% at P7. These data are consistent with the images from an independent lineage tracing study (PMID: 22546693). These data suggest that this study has not obtained the appropriate population of valve mesenchyme cells to use as the terminus for EpiMT.

We thank reviewer #3 for their comments. We hope that the edits incorporated in the revised version of our manuscript have addressed their main concerns.

Major points.

1. Histological data continues to lack quantitative analysis.

We have included quantifications for histological data for mesenchymal cell proliferation (Figure S2H), endothelial-mesenchymal plasticity in endothelial lineages from E9.5-12.5 (Figure S3D-F), endothelial and mesenchymal identity in endothelial lineages from the WT and Sox9 cKO from E9.5-12.5 (Figure S4L), and proliferation in endothelial lineages from WT and Sox9 cKO from E9.5-12.5 (Figure S4N).

2. The authors combined their data with published data (De Soysa et al). They perform integrative analysis but do not show that their analysis overcomes batch effects. In their reply, the authors indicate that the method used, Harmony, is not designed to overlay datasets completely but rather to stitch together datasets in a time series.

There are at least two variables that could account for differences between cells in the different data sets: (1) time point; (2) batch effect. The data sets do not appear to intermingle well (Fig. S1G) and it is not possible to determine if this is due to a difference in the time point or a technical difference due to batch effect. Regardless of the analysis method used, the authors need to provide data that resolves this question.

As mentioned in the initial review, there are many eligible data sets that could be included in the analysis. The authors do not provide a solid reason for selectively using the De Soysa et al data set. Finding similar results in multiple data sets would go far to strengthening the conclusion.

The ability to overcome batch effects when integrating multiple datasets from a time series is a challenging and active research area in computational biology. We have used one of the leading methods for correcting batch effects over timeseries – Harmony – but as the reviewer has mentioned, this does not guarantee that the batch effects are fully eliminated.

There are differences between our datasets and those from De Soysa *et al.* (2019), related predominantly to cell type composition as they were generated from microdissected AVCs versus whole hearts, respectively. However, significant overlap is observed in certain populations that are present across the developmental stages assayed in these datasets, including epicardium, endothelium, and cardiac mesoderm (Figure S1H). Other populations, including early progenitors (mesoderm, neural progenitors, etc.), AV mesenchyme, and STM show little overlap due to their differentiation or emergence during the developmental timepoints assayed or due to the tissues used for library generation.

We are not aware of any purpose-built packages currently available to quantify batch effects across a developmental time series as requested above; however, we have provided additional data which we believe addresses this limitation. We have generated lists of genes specific for each of the batches integrated in this manuscript (Supplementary Table 3). Heatmap visualization shows relatively subtle batch effects present in the integrated data (Supplementary Figures S2A and S5A). Further, gene ontology of these batch-specific genes is dominated by terms related to metabolism and generic biological processes (Supplementary Table 3). We have further included a caveat in the results (lines 105-107), mentioning that some transcriptional differences across batches persist after correction.

We appreciate that there are other cardiac single-cell RNA-seq datasets available besides De Soysa *et al.*; however, most are not suitable as they are composed of cells from later developmental timepoints, from different parts of the heart, or from disease models. Further, older datasets prove difficult to integrate due to low cells numbers and relatively low quality metrics. The data from De Soysa *et al.* (2019) were captured at the correct developmental timepoints to extend our analysis and include a large number of early endothelial and epicardial cells. That being said, we strongly agree that the inclusion of additional datasets would help strengthen our conclusions. We have analyzed data recently published in Feng *et al.* (2022), where they performed single-cell RNA-seq on whole mouse hearts across development from E9.5-P9. Unfortunately, this only captured 837 total endothelial and epicardial-derived cells from E9.5-E12.5 (below). Due to small cell numbers and the capture of cells from the whole heart, only a

couple of cells exhibiting EMP were identified (arrowheads, below), which we did not feel added much impact to our manuscript.

We have further tried to integrate data from human fetal cardiac datasets to show conservation between species, including Cui *et al.* (2019), Sahara *et al.* (2019), and Asp *et al.* (2019). These datasets were generated from whole human fetal hearts but have limited overall numbers of cells (all <4000) and few endocardial and epicardial cells. Further, these data were generated from whole hearts and the stages sequenced are later than the equivalent developmental timepoints we have included from mouse, limiting our overall ability to perform direct comparisons. Due to these constraints, we could not identify rare cells exhibiting EMP (below). For these reasons, we have decided not to include this analysis in the manuscript.

3. The authors also only obtained a single replicate of the *Sox9* cKO samples so reproducibility was not demonstrated. The authors replied that the one library was composed of multiple embryos from two litters and that other manuscripts are published with data sets that do not show reproducibility. Multiple embryos in one library means the library represents the average of multiple embryos but does not demonstrate reproducibility. Low scientific rigor in other manuscripts is not good justification for low rigor.

We agree that an additional replicate from the *Sox9* cKO would be ideal. Obtaining tissue for sequencing from the *Sox9* cKO embryos was difficult. Generating this single-cell library required simultaneous dissections, genotyping, dissociations, and library preparations by several members of my lab. This presented unique logistical challenges during the initial library generation and is largely the reason only a single replicate was made. At this time, to generate new embryonic libraries, we would further need to expand our mouse colony. This colony is difficult to manage due to the health of the mice and the complicated genetics of the necessary breeding strategy to obtain embryos with the desired genotype. Unfortunately, this would require several months to a year to accomplish, which is beyond the stated editorial timeframe.

To address this issue, we have validated our findings around the *Sox9* cKO using histology, image quantifications, and flow cytometry. These validations provide additional evidence of the changes in the proportions of endothelial, mesenchymal, and hybrid populations in the *Sox9* cKO AVC we observed by transcriptomics. We appreciate that this compromise is less than ideal but feel that the validation supports our conclusions.

4. In the initial review we requested improved clarity on how differentiation potential and EMP were defined and calculated. Although the authors indicated in their reply that the information is included in the revised methods, these terms remain opaque. Moreover, there is no validation that these scores are meaningful. In some cases, EMP is indicated as a binary value, implying use of a threshold. How the threshold was established and validated is also not clearly stated. Does this threshold differ between EpiMT and EndMT?

Expanded information related to the definitions of differentiation potential and EMP has been included in the main text and methods sections of the manuscript. Definitions related to differentiation potential are included in lines 128-132 and definitions related to EMP can be found in lines 582-594.

Differentiation potential, as presented in Palantir (Setty *et al.* 2019), is a quantification of plasticity of a cell, where multipotent cells have the highest differentiation potential and mature terminal states have the lowest potential. As differentiation is asynchronous, sequencing a population of differentiating cells yields a snapshot representing a range of cell states. Palantir uses single-cell transcriptomic data to pseudotime order cells in a developmental process, then assigns a probability to each cell along this continuum to differentiate to defined terminal states.

EMP is generally defined as the strong co-expression of factors associated with mesenchymal and epithelial identity. We calculated gene profiles for endothelial, epicardial, and mesenchymal identity using annotated lists of biomarkers and master regulators. Conservative minimum thresholds in each gene profile for meeting endothelial, epicardial, or mesenchymal identity were determined (Figures S3A and S6A). Cells that met the threshold for both epithelial and mesenchymal were considered to be exhibiting EMP. While EMP represents a continuum, we provided these data as a binary for easy visualization on the heatmap.

We agree that the validation of differentiation potential and the plasticity of cells exhibiting EMP is a priority. This would require the transgenic labelling of these hybrid cells and tracing their contribution to endocardial and mesenchymal populations during development. We are currently in the planning stages of generating dual recombinase reporter lines, where dual activity of endothelial and mesenchymal-specific recombinases would indelibly mark the descendants of hybrid cells in the AVC. However, this requires a great deal more work and we feel is more suited as a separate manuscript. We have addressed this caveat in the discussion (lines 426-429) and have specifically mentioned plans to validate this down the road.

5. The framework of the epicardial trajectory analysis remains confusing. Consistent with literature, the authors state “AV EPDCs are specified from epicardium, and not directly from a mesodermal progenitor earlier in development.” If this is the case, then why is the trajectory analysis set up so that AV EPDCs are alternate termini from an STM progenitor start point, rather than a sequence of STM progenitor -> epicardium -> AV EPDC?

We apologize for the confusion here. Indeed, the trajectory analysis is set up such that the AV EPDC trajectory passes through an epicardial intermediate. We have clarified this through the

addition of a separate diffusion maps illustrating the trajectory taken by the cells (Supplementary Figures S2I and S5D) during differentiation. We have also explicitly labelled the epicardial intermediate state in Figure 5D as epicardial cells progress towards an AV EPDC identity.

6. The authors specified AV mesenchyme as a terminus of epicardial differentiation. It seems this is the same AV mesenchyme used for analysis of EndoMT -- **making the assumption that both EndoMT and EpiMT yield similar valve mesenchyme cells**, as opposed to distinct subsets of valve mesenchyme cells. What is the basis for this assumption? As described in PMID 22546693 and 30111655, EpiMT contributes to only a small fraction (~5%) of AV valve mesenchyme at this stage.

This is an important caveat, which we have addressed in both the results and the discussion portions of the manuscript. We have specifically stated that lineage contribution remains difficult to pinpoint and that the inclusion of mesenchymal cells in the EpiMT datasets specifically acts as more as an anchor point, providing developmental context for us to investigate the process of EpiMT (lines 271-274).

Interestingly, we could not find clear transcriptional differences between endocardial- and epicardial-derived mesenchymal populations. We incorporated lineage tracing (*Tie2-cre;tdTomato*) as a way to differentiate these two populations in our single-cell datasets at E12.5. While histologically, tdTomato protein expression in these embryos is specific to endothelial-derived lineages (example below), we were surprised to find that *tdTomato* expression in the scRNA-seq data was weak, diffuse, and not specific to a single population of mesenchymal cells. We subsequently could not use *tdTomato* expression as a way to resolve lineage contribution.

We subsequently looked at other heart valve datasets, including Hulin *et al.* (2019), which analyzed postnatal development of the mouse heart valves. They identify a population of valve interstitial cells that share some transcriptional parallels with epicardial cells and that may be derived by EpiMT from the epicardium earlier in development. Unfortunately, in our datasets, genes specific to this population are either not present (e.g. *C4b*) or widespread within epicardial and mesenchymal populations (e.g. *Tcf21*).

This remains an important and interesting question, which would likely require a complex lineage tracing strategy (potentially dual recombinase) along with FACS to segregate endothelial and epicardial-derived populations prior to single-cell library preparation. Alternatively, this could be addressed using different genomics methods, like scATAC-seq, to identify developmental legacy of AV mesenchyme during development. Unfortunately, this remains beyond the scope of the current manuscript, though we have included these future directions in the discussion (lines 431-432).

7. In the revised manuscript, the authors claim identification of a “unique AV EPDC population”. The label “unique” is used multiple times to describe this population. What is the meaning of

“unique” – unique from what? The authors identify some markers that distinguish chamber and AV EPDCs (Fig. S5H-I), but it appears the main marker used, Hic1-nLacZ, does not distinguish these types of EPDCs (Fig. 5C).

We have removed claims of uniqueness in the manuscript related to the AV EPDC population. While we did identify markers specific to chamber and AV EPDCs in the single-cell data, our attempts to validate these targets with histology were unsuccessful due to poor signal and high background.

8. Since the authors declined to validate the cell signaling analysis, particularly the prediction that AV mesenchyme and endocardium ligands contribute most to EpiMT, this should be moved to the discussion and stated with the caveat that it has not been experimentally validated.

We have removed this statement from the results.

REVIEWERS' COMMENTS

Reviewer #3 (Remarks to the Author):

No further comments.